# Interpreting learned search: finding a transition model and value function in an RNN that plays Sokoban

## Abstract

We partially reverse-engineer a convolutional recurrent neural network (RNN) trained to play the puzzle game Sokoban with model-free reinforcement learning. Prior work found that this network solves more levels with more test-time compute. Our analysis reveals several mechanisms analogous to components of classic bidirectional search. For each square, the RNN represents its plan in the activations of channels associated with specific directions. These state-action activations are analogous to a *value function* – their magnitudes determine when to backtrack and which plan branch survives pruning. Specialized kernels extend these activations (containing plan and value) forward and backward to create paths, forming a *transition model*. The algorithm is also *unlike* classical search in some ways. State representation is not unified; instead, the network considers each box separately. Each layer has its own plan representation and value function, increasing search depth. Far from being inscrutable, the mechanisms leveraging test-time compute learned in this network by model-free training can be understood in familiar terms.

## 1 Introduction

Traditional online planning algorithms such as alpha-beta or Monte Carlo Tree Search (MCTS) attempt to accomplish a goal by exploring many possible courses of action (plans) using a *transition model* [53]. These algorithms can use additional compute to improve decisions by increasing the number of plans evaluated or the length of considered plans (the planning horizon). At each environment step, the algorithm considers many plans, ranks each according to its outcome, and picks the first action of the best plan. Often, the goal is further away than the horizon, so the outcome of intermediate states at the horizon must be evaluated with an approximate *value function*. To consider fewer plans (and thus be able to search deeper), these algorithms use move generation *heuristics* to simplify the problem and avoid exploring some actions[1].

It is difficult to craft heuristics and value functions for complex environments, leading to work such as AlphaGo and AlphaZero that combines MCTS with machine-learned evaluation and move generation [60, 61, 11]. This hybrid approach uses the model for high-quality move generation and evaluation, while the search backbone uses extra compute to improve performance via more and deeper exploration. Recent work has shown that large language models (LLMs) exhibit *test-time scaling*: using more compute can generate better answers [45, 13]. However, unlike previous examples, it is unclear exactly *how* this additional compute is used to improve performance.

How *does* test-time scaling work? We study a model organism for test-time scaling: a Deep Repeating ConvLSTM (DRC) trained to play the Sokoban puzzle game. [22, 62]. We focus on the DRC because

---

[1]In alpha-beta search, this corresponds to trying better moves first so branches can be pruned later on by the $\beta$-threshold [53], and in some MCTS variants, this corresponds to the prior policy [61].

Submitted to 39th Conference on Neural Information Processing Systems (NeurIPS 2025). Do not distribute.

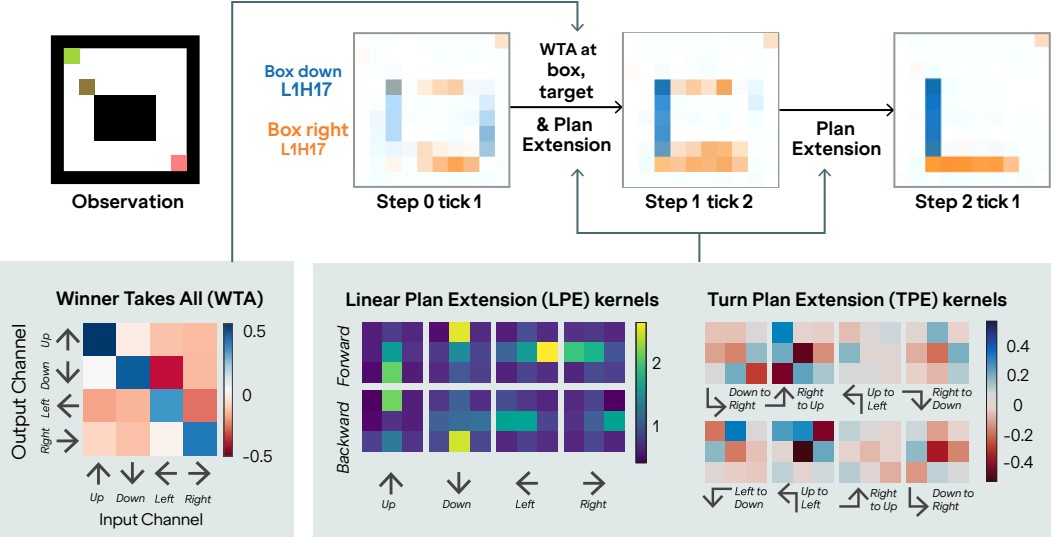

Figure 1: A situation with two equally good paths from the box ▮ to the target ▮. The sum of box-down (L1H17) and box-right (L1H13) channels shows that the network searches forward from the box and backward from the target. Both paths (down-then-right and right-then-down) are visible at step 0 tick 1 (left) due to the encoder; and the down and right channels have similar activations on the box square (gray). From step 0 tick 1 until step 2 tick 1 (Section 3.1 defines 'tick'), the plans are extended in the same direction by Linear Plan Extension (LPE) kernels (bottom-middle) and extended into switching directions by Turn Plan Extension kernels (bottom-right), stopping (Figure 6) on signals corresponding to reaching the target or hitting obstacles. The plan at the box square is resolved at step 1 tick 2 using a Winner-Takes-All (WTA) mechanism. The average WTA kernel weights (bottom-left) subtract each channel from all the others, which through a sigmoid approximates an argmax. The magnitude of the diagonal entries (stronger for down than right) break ties.

previous work has established that it benefits from test-time compute while being a small enough network to make intensive mechanistic interpretability tractable. Additionally, actions and goals in Sokoban are concrete and observable, in contrast to the unclear goals and state-action space of LLMs.[2]

Our primary contribution is to partially reverse-engineer the algorithm that the DRC learned. To the best of our knowledge, this work advances the Pareto frontier between the completeness of a mechanistic explanation and the complexity of the phenomenon: we characterize more of the Sokoban algorithm than any previous work [2, 62] that has aimed to do so, and our work is more complete or more complex than any other related work (see Section 6) in mechanistic interpretability. We find that the DRC contains several analogues to classic online search, performing bidirectional search as argued by Bush et al. [2]. To explain the algorithm, we first focus on the data representation.

**Representation.** Since the DRC repeats the same computations for each square with limited feedforward depth, it cannot use more "memory" for longer paths. However, its convolutional structure allows it to use a distributed representation: at each square, the activations of agent and box direction *path channels* (Section 4) encode the direction to go from that square. The DRC uses short- and long-term path channels to represent going in different directions at different times the square is visited. A path is represented by adjacent squares having path channel activations that indicate a move to the next square in the sequence. How are these path channel activations constructed?

**Algorithm.** These *path channel* activations are initialized by the encoder kernels to begin plan segments at the agent, boxes, and targets (Section 5.1). The plan segments are then extended bidirectionally by forward and backward plan extension kernels which extend them linearly or with a turn until an obstacle is met, functioning as a *transition function* that extends plans with valid next actions (Section 5.2). These plan extension kernels also double to propagate negative activations

---

[2]Tokens are concrete, but LLM goals and world models likely only make sense at a higher level of abstraction.

bidirectionally, pruning unpromising paths (Section 5.3). A winner-takes-all (WTA) mechanism strengthens the strongest directions in the path channel activations and inhibits the weaker directions which, along with the sigmoid nonlinearities, causes the plan segments to commit to the higher activation directions when there are multiple options available (Section 5.3).

Taken together, the backtracking and WTA mechanisms show that the path channel activations are analogous to a *value function*, propagating positive and negative value information along a path, and being used to select high value subplans. This provides a mechanism for specialized *heuristics* to influence the final plan: simply add or subtract activation to strengthen or inhibit a path.

**(Dis)analogies to classical algorithms.** Thus, the DRC has analogues to key components of classical algorithms in the form of a plan representation in the path channel activations, which are repurposed as a valuation mechanism, and constructed according to a transition function. However, the convolutional computational structure imposes subtle differences. The plan's representation is distributed across activations for each square, leading to potential inconsistencies that need to be suppressed by the WTA mechanism. These activations are used as a value function, but they propagate subplan value information along the path via the convolutional plan extension kernels so that the effective value of the plan is the result of an equilibrium rather than a variable assignment.

## 2 Background

**Mechanistic interpretability.** Linear probing and PCA have been widely successful in finding representations of spatial information [66] or state representations and game-specific concepts in games like Maze [27, 31, 39], Othello [32, 41], and chess [36, 57, 29]. However, these works are limited to input feature attribution and concept representation, and do not analyze the algorithm learned by the network. Recent work has sought to go beyond representations and understand key circuits in agents. It is inspired by earlier work in convolutional image models [5] discovering the circuits responsible for computing key features like edges, curves, and spatial frequency [4, 43, 56]. In particular, recent work has found mechanistic evidence for few-step lookahead in superhuman chess networks [28, 57], and future token predictions in LLMs on tasks like poetry and simple block stacking [33, 37, 47]. However, these works still focus on particular mechanisms in the network rather than a comprehensive understanding of the learned algorithm.

**Planning in Sokoban.** Sokoban is a grid-based puzzle game with walls ■, floors ▢, movable boxes ■, and target tiles ■ where the goal of the agent ■ is to push all boxes onto target tiles. Since boxes can only be pushed (not pulled), some wrong moves can make the puzzle unsolvable, making Sokoban a challenging game that is PSPACE-complete [12], requires long-term planning, and a popular planning benchmark [48, 51, 23]. Guez et al. [22] introduced the DRC architecture family and showed that $DRC(3, 3)$ achieves state-of-the-art performance on Sokoban amongst model-free RL approaches and rival model-based agents like MuZero [55, 9]. They argue that the network exhibits *planning* behavior since it is data-efficient in training, generalizes to multiple boxes, and benefits from additional compute. Specifically, the solve rate of the DRC improves by 4.7% when the network is given extra thinking time by feeding in the first observation ten times during inference. Bush et al. [3] use logistic regression probes to find a causal representation of the plan in the DRC, which improves with compute, and speculate that it might be performing bidirectional search. Taufeeque et al. [62] find that training incentivizes the DRC to often wait for a few steps before acting and during those waiting steps the plan changes more quickly, indicating the policy has a meta-strategy of seeking test-time compute when needed.

## 3 Methodology

### 3.1 Network architecture

We analyze the open-source $DRC(3, 3)$ network trained by Taufeeque et al. [62] to solve Sokoban, who closely followed the training setup of Guez et al. [22]. The network consists of a convolutional encoder, a stack of 3 ConvLSTM layers, and an MLP head for policy and value function (in the sense of the RL policy training, not path valuation) prediction. Each ConvLSTM block perform 3 ticks of recurrent computation per step. The encoder block $E$ consists of two $4 \times 4$ convolutional layers

without nonlinearity, which process the $H \times W \times 3$ RGB observation $x_t$ into an $H \times W \times C$ output $e_t$ with height $H$, width $W$, and $C$ channels, at environment step $t$.

Figure 2 visualizes the computation of the ConvLSTM layer. Each of the ConvLSTM layers at depth $d$ and tick $n$ in the DRC maintains hidden states $h_d^n, c_d^n$ with dimensions $H \times W \times C$ and takes as input the encoder output $e_t$, the previous layer's hidden state $h_{d-1}^n, c_d^{n-1}$, and its own hidden state $h_d^{n-1}$ from the previous step. The ConvLSTM layer computes four parallel gates $i, j, f, o$ using convolutional operations with $3 \times 3$ kernels that are combined to update the hidden state. For the first ConvLSTM layer ($d = 0$), the architecture uses the top-down skip connection from the last ConvLSTM layer ($d = 2$). This gives the network $3 \cdot 3 = 9$ layers of sequential computation to determine the next action at each step. The final layer's hidden state

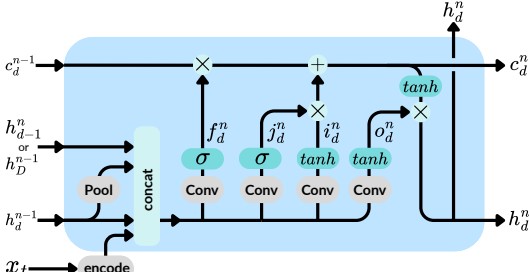

Figure 2: The ConvLSTM block in the DRC. Note the use of convolutions instead of linear layers, and the last layer of the previous tick ($h_D^{n-1}$) as input to the first layer. "Pool" refers to a weighted combination of mean- and max-pooling.

at the last tick is processed through an MLP head to predict the next action and value function. The DRC$(3, 3)$ architecture is shown in Figure 10, with additional architecture, training, and dataset details provided in Appendices B and C. Unless mentioned otherwise, the dataset of levels for all results is the medium-difficulty validation set from Boxoban [21].

**Notation** Each DRC tick ($n = 0, 1, 2$) involves three ConvLSTM layers ($d = 0, 1, 2$), each providing six 32-channel tensors ($h_d, c_d, i_d, j_d, f_d, o_d$). Channel $c$ of tensor $v_d$ is denoted L$d$V$c$.

## 3.2 Interpretability Techniques

This paper employs the following techniques from the mechanistic interpretability literature [17, 43, 59] to reverse-engineer the planning algorithm of the DRC$(3, 3)$ network. The first four help us form hypotheses (denoted [H]) about the DRC, and the last two help us test them (denoted [T]). Through the synthesis of these techniques, we build a mechanistic account of how the DRC$(3, 3)$ network represents, searches for, refines, and executes plans to solve Sokoban puzzles.

**[H] Feature Identification and Label.** Interpretability requires identifying legible features that the network uses to make decisions [30, 43]. We analyze individual channels of the hidden state $h$, seeking to label all channels by their purpose. Wherever necessary to explain a mechanism, we further decompose the activations of the hidden state channels into the components received from the $c, i, j, f, o$ gates based on Equations (7) and (8). To label channels, we observe them, form hypotheses about their purpose, and test these using a *test set*, *causal interventions* or *ablation*.

**[H] Kernel Visualization.** After identifying what the channel activations represent, we visualize the convolutional kernels connecting various input channels in the network to intuitively understand the mechanisms (or "circuit") involved in computing an output channel [64, 4].

**[H] Encoder Simplification.** Individually, the encoder weights have no privileged basis [16]. To interpret the weights of the encoder, we use the associativity of linear operations to combine the convolution kernels of the encoder and the $i, j, f, o$ gate weights that process the encoder output $e_t$ into a single convolutional layer. This results in $9 \times 9$ convolution kernels directly mapping observations to each gate (Figures 4 and 22).

**[H] Direct effect.** To study which inputs channels contribute the most to an output channel gate, we sort and filter the input channels by their direct effect, computed as the largest magnitude of activation added to the output across all squares in the grid.

**[T] Causal Intervention.** To ground our interpretation of the activations and weights in the network's behavior, we intervene on the activations [32, 19, 68] and weights of the network and observe whether the network's behavior changes as expected based on the intervention $\mathbf{x}' \leftarrow \alpha \cdot \mathbf{x} + \mathbf{c}$,

where **x** can represent the activations or weights depending on the experiment, $\alpha$ is a constant multiplier, and **c** is a constant steering vector [63, 52].

**[T] Ablation.** Ablation is one specific causal intervention technique used to 'remove' components from a neural network and thus understand their importance [68, 65, 10]. We perform mean-ablation on the activations as $\mathbf{x}' \leftarrow \mathbb{E}[\mathbf{x}]$ replacing the activations of a component with its mean over some episode distribution, and measure the drop in performance to decide which components to focus on in our analysis. We also perform zero-ablation on the *kernel weights* by replacing a set of kernel weights with zero to validate our interpretation of the kernels [38].

# 4 The Plan Representation

At each layer, the DRC has $C$ channels, each of which is a $H \times W$ array. The DRC repeats the same computations convolutionally over each square. This results in a subset of channels representing the plan where each channel corresponds to a movement direction. If the agent or a box is at a position where the channel is activated, this causes the DRC to choose that channel's direction as the action (Figure 3, left).

Table 1: Channel groups, their definitions and counts for each direction (up, down, left, right).

| Group | Definitions | Channels |
|---|---|---|
| Box-movement | Path of box (short- and long-term) | 20 (3, 6, 5, 6) |
| Agent-movement | Path of agent (short- and long-term) | 10 (3, 2, 1, 4) |
| Grid Next Action (GNA) | Immediate next action, represented at agent square | 4 (1, 1, 1, 1) |
| Pooled Next Action (PNA) | Pools GNA to represent next action in all squares | 4 (1, 1, 1, 1) |
| Entity | Target, agent, or box locations | 8 |
| Combined path | Aggregate 2+ directions from movement channels | 29 |
| No label | Difficult to interpret channels | 21 |

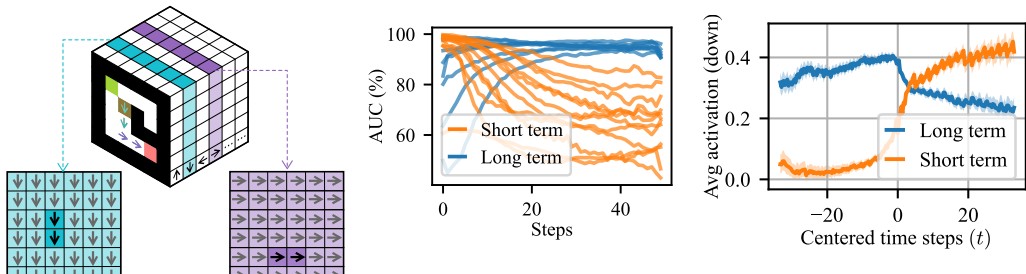

Figure 3: **Left**: illustration of path channels. Each channel is a 2D slice of the 3D activations, which activates highly at a square to indicate the direction it is associated with. **Middle**: The network represents actions at different horizons separately to express concepts like "first time go right, second time go down." Short-term box-movement channels accurately predict ($> 95\%$) the next 10 steps of box-movement. For later box-movements, long-term channels represent actions accurately, with AUC approaching $> 95\%$. **Right:** Average activations of short-term (L0H2, L1H17, L1H19) and long-term (L0H14, L0H20, L1H14) box-down channels averaged across squares where a non-down action is taken at the centered time step $t = 0$ and a down action is taken at $t > 0$. The down action is stored in the long-term channel at $t < 0$ and transferred into the short-term channels after the non-down action occurs and down becomes the next action to take at that square.

Manual inspection of every channel across all layers revealed that most channels are interpretable (Table 1). Detailed labels are in Tables 7 and 8 of the Appendix. We group the channels into seven categories: 1) box-movement and 2) agent-movement channels that, for each cardinal direction, activate highly on a square in the grid if the box or agent moves in that direction from that square at a future timestep. The probes from Taufeeque et al. [62] and Bush et al. [3] aggregated information from these channels. 3) The combined path channels that aggregate various directions from the box- and agent-movement channels. 4) The GNA channels that extract the next action from the previous groups of channels. 5) The PNA channels that pool the GNA channels which are then picked up by the

MLP to predict the next action. 6) The entity channels that predominantly represent target ▨ locations, with some also representing box ▨ and agent ▨ locations, and 7) Some channels we understand very little of ('no-label'). We define the *path channels* as the set comprising the box-movement, agent-movement, and combined path categories, as they collectively maintain the complete plan of action for the agent. The remaining groups (GNA, PNA, entity, and no-label) are termed *non-path channels*, storing primarily short-term information, with some state for move selection heuristics. The box- and agent-movement channels further decompose into short- and long-term channels (Table 7, Appendix L). As illustrated in Figure 3 (middle), these channels collectively predict future movements up to 50 steps ahead with high accuracy (AUC $> 95\%$, area under the ROC curve). Figure 14 shows the AUC curves for each channel separately.

**Long-term path channels.** The network utilizes the *long-term* channels to manage spatially overlapping plans for different boxes intended for different times. Figure 3 (right) illustrates this: in cases where two boxes pass through the same square sequentially in different directions with the first box moving at $t = 0$, the long-term channel for the *second* move activates well in advance ($t \ll 0$), while its corresponding short-term channel only becomes active after the first move is completed ($t = 0$). Figure 16 shows the mechanism of this transfer is primarily mediated through the $j$-gate.

**Ablating the state.** To test whether non-path channels hold state, we performed a single-step cache ablation. This ablation replaces the activations of a target group of channels with the hidden state generated from running the policy on the previous observation starting from a zero state, effectively removing long-term dependencies while preserving short-term computations within those channels. Intervening on the 59 path channels caused a substantial 57.6% drop in the solve rate. By contrast, intervening on the 37 non-path channels resulted in a 10.5% performance decrease ((a significantly smaller, yet non-negligible, decrease). Controlling for channel count, intervening on a random subset of 37 path channels still led to a 41.3% drop in solve rate. This evidence strongly suggests that the computations essential for long-term planning on difficult levels are primarily carried out within the identified path channels.

**Uncharted behaviors and channels.** The DRC does more things that we do not yet understand. For example, the plan extension has a tendency to move towards boxes and targets, as opposed to exploring every possible direction, but only when the box and target are at most 10-15 squares apart. There are many channels for which we do not have labels, though we are confident that these channels only affect the action through the short-term path channels because several short-term channels have $> 99\%$ AUC (Figure 3) for predicting the next action. However, their activations are sometimes important and they appear to be used on more difficult levels, so we call these channels *heuristics*.

**Causal intervention** We verify the channel labels by performing causal interventions on the channels. We modify the channel activations based on their labels to make the agent take a different action than the one originally predicted by the network. We collect a dataset of 10,000 transitions by running the network on the Boxoban levels [21], measuring the fraction of transitions where the intervention succeeds at causing the agent to take any alternate target action, following the approach of Taufeeque et al. [62]. Table 2 shows high intervention scores for every group except the agent-movement channels. The lower score for agent-movement channels is because they are causally relevant only when the agent is not pushing a box, a condition we did not filter for. We also compare our results to probes trained by Taufeeque et al. [62] to predict box and agent movements and find that intervening based on our channel labels is more effective than using their probes. In Appendix E, we further validate our channel labels.

Table 2: Causal intervention scores for different channel groups, alongside comparative probe scores from Taufeeque et al. [62].

| Group | Score (%) |
| --- | --- |
| Pooled Next Action (PNA) | $99.7 \pm 0.2$ |
| Grid Next Action (GNA) | $98.9 \pm 0.4$ |
| Box- and agent-movement | $88.1 \pm 1.9$ |
| Box-movement | $86.3 \pm 2.1$ |
| Agent-movement | $53.2 \pm 2.1$ |
| Probe: box movement | $82.5 \pm 2.5$ |
| Probe: agent movement | $20.7 \pm 0.7$ |

We thus conclude that the network's plan resides primarily within the identified box-movement and agent-movement channels, which are mapped to the next action through the GNA/PNA channels. In Appendix F, we explain the mechanisms that map these plans to the next actions.

## 5 The Planning Algorithm

Bush et al. [2] observed qualitatively with their box-movement probe that the DRC(3, 3) network forms plans by forward chaining from boxes and backward chaining from targets in parallel in the first few steps. We find concrete evidence of this in the weights of the network by analyzing the kernels associated with the path channels.

### 5.1 Initializing the plan

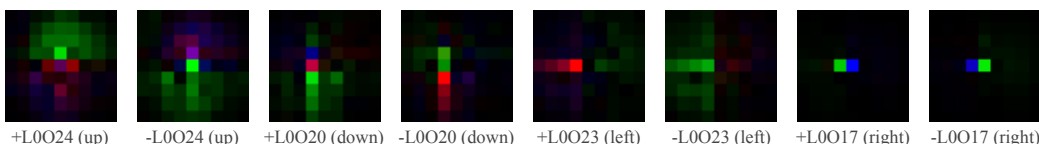

+L0O24 (up)  -L0O24 (up)  +L0O20 (down)  -L0O20 (down)  +L0O23 (left)  -L0O23 (left)  +L0O17 (right)  -L0O17 (right)

Figure 4: Visualizations of combined kernels that map from the RGB input to the *o*-gate of the up, down, left, and right box-movement channels of layer 0. The negative and positive RGB components are visualized separately. The kernels activate squares along (for agent ■ and box ■) and against (for target ■) the channel's direction to initialize the forward and backward plan chains, respectively. The kernel for L0O17 (right) initializes plan chains only on the agent and box square.

Analysis of the combined encoder kernels mapping to box-movement channels (Figure 4) reveals structures that initiate planning. These kernels detect relevant features—such as targets, boxes or the agent's position—a few squares away along (for box) or against (for target) the channel's specified cardinal direction. This allows the network to activate initial plan segments, effectively starting the bidirectional search.

### 5.2 The Transition Model

The DRC's kernels contain convolutions analogous to the transition model of a classical planning algorithm because they encode how the environment changes in response to the agent's actions or how to reach a state. They strongly bias the plan expansion process towards valid state transitions.

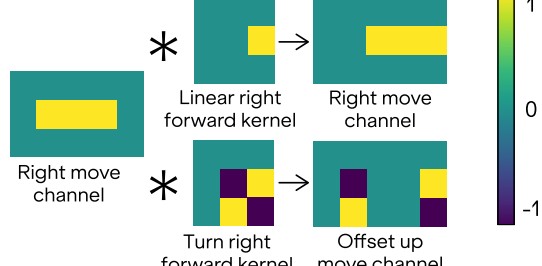

**Plan Extension Kernels.**   While encoder kernels initiate plan fragments, connecting these forward and backward chains requires an extension mechanism operating in the recurrent hidden state. This is accomplished by specialized "plan-extension kernels" within the recurrent weight matrices.

*Linear plan extension (LPE) kernels* (see also Figure 1) propagate the plan linearly, extending

Figure 5: The effect of convolving a movement channel with the plan extension kernels. Both forward and backward turns happen with the same turn kernel. The up channel is spatially offset (1 square right and down) to place turn activation at the correct square.

it one square at a time along the channel direction label. Separate kernels exist to facilitate both forward chaining from boxes and backward chaining from targets. *Turn Plan Extension (TPE) kernels* (see also Figure 1) switch activations from one channel to another channel that represents a different direction. The LPE kernels have larger weight magnitudes compared to the TPE kernels, thus encoding agent's preference to take turns only when linear plan extension stops expanding along a direction.In Appendix M, we demonstrate that weight steering based on our insights into the plan extension mechanism can help solve larger adversarial levels previously identified by Taufeeque et al. [62].

These kernels constitute a transition model in the sense that they encode the dynamics of Sokoban. If the agent or box moves in a direction, then the adjacent square in that direction is activated, with a default of continuing in the same direction.

**Stopping Plan Extension.** Plan extension does not continue indefinitely. It must stop at appropriate boundaries like targets, squares adjacent to boxes, or walls. We observe (Figure 6) that this stopping mechanism is implemented via negative contributions to the path channels at relevant locations. These stopping signals originate from either the encoder or hidden state channels that represent static environmental features (such as those in the 'entity' channel group, Table 7), effectively preventing the plan from extending beyond targets or into obstacles.This aspect of the transition model prevents the DRC from adding impossible transitions to its path.

**State transitions.** In Appendix G, we show the mechanisms that update the plan representation on state transitions, solidifying the notion that the DRC has an internal transition model.

### 5.3 The Valuation Mechanism

The value function is a key component of classical planning algorithms. We now argue that the *activations* of the path channels are used analogously to a value function: aggregating and propagating reward information about a path, and being used to select high-value plans.

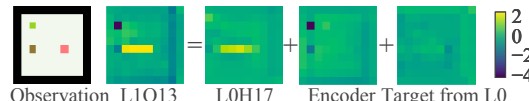

Observation  L1O13  L0H17  Encoder  Target from L0

Figure 6: Plan stopping mechanism demonstration shown through *o*-gate contributions of the box-right channel (L1H13). The direct effect shows that convolving the forward and backward right-plan-extension kernel on the converged box-right channel (L0H17) spills into the squares of the box and the target. The encoder and the target channels from layer 0 add a negative contribution to counteract the spillover and stop the plan extension.

**Backtracking mechanism.** The plan extension kernels are cleverly repurposed to allow the algorithm to backtrack from bad paths. As part of its bidirectional planning, the DRC has forward and backward plan extension kernels, so negative activations at the end of a path are propagated to the beginning by the backward kernel, and negative activations at the beginning of a path are propagated to the end by the forward kernel. This allows the DRC to propagate negative activations along a path, thus pruning unpromising path fragments. See Appendix I for an example.

This is analogous to backward chaining in a classical game tree expansion algorithm, in that it propagates negative reward information backward from invalid or low-reward paths. However, it is somewhat more complex: rather than every path channel in a single plan having the same activation, the activation propagates forward and backward along the path using the plan extension kernels.

**Bidirectional planning.** This allows the DRC to construct paths using something like a standard bidirectional search algorithm – plan fragments get extended by the transition model in the plan extension kernels, stopped by obstacles, and backtracked entirely to prune bad branches. But how does it stitch the fragments together into a consistent, high-value plan?

**Winner-takes-all mechanism.** To select a single path for a box when multiple options exist, the network employs a Winner-Takes-All (WTA) mechanism among *short-term* path channels. Excluding the *long-term* path channels allows the DRC to maintain plans for later execution without inhibiting them. Figure 1 (bottom-left) shows that weights connecting path channels for various directions cause the path channel activations to inhibit each other at the same square. The direction with the strongest activation suppresses activations in alternative direc-

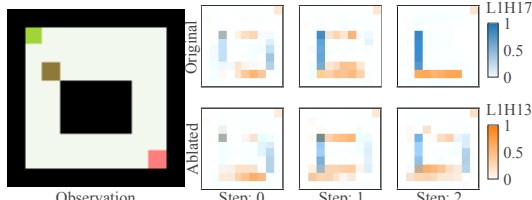

Figure 7: After zero-ablating the kernels connecting the box-down and box-right channels, the WTA mechanism cannot suppress the right-down plan.

tions which, combined with the sigmoid activation, ensures that only one direction's path channels remain active for imminent execution. We construct a level with equally viable paths to causally demonstrate (Figure 7): initially, both paths have similar activations, but the slightly stronger one quickly dominates in steps 1 and 2 and deactivates the other via this inhibitory interaction. Zero-ablating the kernels between the channels of the two directions eliminates the WTA effect, leaving both potential paths simultaneously active. Thus, we conclude that kernels connecting various short-term box-movement path channels implement this crucial selection mechanism.

**Path channel activations as a value function analogue.** These findings show that not only do the path channel activations *represent* the plan, they are analogues to the DRC's value function. The plan extension kernels propagate and aggregate value information forward and backward along a path. The WTA mechanism ensures that the highest value plans are chosen, and cause the path to connect to higher value segments when connecting bidirectional plan segments. This distributed representation works with the DRC's convolutional architecture and allows the repeated application of the same computations at each square to propagate value information through the constructed paths.

# 6 Related work and discussion

**Mechanistic explanations.** To the best of our knowledge, our work advances the Pareto frontier of the complexity of a neural network, versus the level of detail in the description of its mechanism. Much work focuses on the mechanisms of large language models. LLMs are more complex than the DRC, but the algorithms these papers explain are simpler as measured by the size of the abstract causal graph [19, 6]. Examples include work on GPT-2 small [65, 24, 15], Gemma 2-2B [34, 42], Claude 3.5 Haiku [33, 35], and others [70]. A possible exception is Lindsey et al. [33], which contains many simple explanations whose graphs together would add up to a graph larger than that of the present work. However, their explanations rely only on empirical causal effects and are local (only valid in their prompt), contrasting with weight-level analysis that applies to all inputs. Pioneering work in understanding vision models [43, 56, 64] is very thorough in labeling features but provides a weight-level explanation for only a small part of InceptionV1 [4]. Other work focuses on tiny toy models and explains their mechanisms very thoroughly, such as in modular addition [40, 8, 69, 50, 20, 67], binary addition [49], small language transformers [44, 25], or a transformer that finds paths in small binary trees [1].

**DRC in Sokoban.** Taufeeque et al. [62] and Bush et al. [3] find internal plan representations in the DRC by predicting future box and agent moves from its activations using logistic regression probes. Some of their probes are causal, others can be used to generalize the DRC to larger levels; however, further analysis is primarily based on qualitative probe and model behavior rather than mechanisms.

**Mesa-optimizers.** Hubinger et al. [26] introduced the concept of a *mesa-optimizer*, an AI that learns to pursue goals via internal reasoning. Examples of mesa-optimizers did not exist at the time, so subsequent work studied the problem of whether the learned goal could differ from the training signal, *reward misgeneralization* [14, 58]. Oswald et al. [46] argued that transformers do in-context linear regression and are thus mesa-optimizing the linear regression loss, which is hardly agentic behavior. Modern AI agents appear to reason, but whether they internally optimize a goal is unresolved.

The present work answers *agentic mesa-optimizer existence* in the affirmative. We present a mesa-optimizer, then point to its internal planning process and to its learned value function. The value function differs from what it should be from the training reward, in benign ways: the training reward has a $-0.1$ per-step term, but the value encoded in the path channels *do not* capture plan length at all. In fact, which path the DRC picks is a function of which one connects to the target first, encoding the preference for shorter paths purely in the LPE and TPE kernels (Appendix J). To compute the value head (critic), the DRC likely counts how many squares are active in the path channels.

# 7 Conclusion

This partial reverse-engineering shows that, while the DRC develops several analogues of components of classical planning such as a plan representation, transition function, and value function, its implementation is deeply influenced by its convolutional structure and is characterized by frequent reuse. The plan is represented as activations in path channels for each square, which are repurposed as a valuation mechanism. Plan extension kernels extend plan fragments bidirectionally, while propagating value information along the path. The winner-takes-all mechanism stabilizes the plan into taking single actions at a time at each square, and chooses the highest-value subplan segments. The DRC does everything everywhere all at once, implementing familiar mechanisms in alien ways.

We were able to understand a planning algorithm, which was learned completely model-free, in familiar terms. This raises the hope that, if LLM agents are internally performing search, it is possible to find, audit, and correct their goals.

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

# Appendix

## A    Common components of search algorithmns

A search algorithm requires three key components:

1. A representation of states
2. A transition model that defines which nodes (states) are reachable from a currently expanded node when taking a certain action
3. A heuristic function that determines which nodes to expand

The heuristic varies by algorithm:

- For A*, it is distance$(n)$ + heuristic$(n)$ [53]
- For iterative-deepening alpha-beta search (as used in Stockfish), the heuristic comprises move ordering and pruning criteria [7]
- For AlphaZero/MuZero MCTS, it uses the UCT formula pre-rollout, incorporating backed-up value functions and a policy with Dirichlet noise [61, 55]

In all cases, the expansion process influences the relative evaluation of actions in the starting state. The final action selection relies on a value function:

- A*: Uses the actual path distance when plans have been fully expanded [53]
- AlphaZero/MuZero MCTS: Employs backpropagated estimated values combining rollout and final score [61, 55]
- Stockfish 16+: Utilizes the machine-learned evaluation function at leaf nodes [7]

## B    Network architecture

The DRC architecture consists of an convolutional encoder without any non-linearities, followed by $D$ ConvLSTM layers that are repeated $N$ times per environment step, and an MLP block that maps the final layer's hidden state to the value function and action policy.

For all $d > 1$, the ConvLSTM layer updates the hidden state at each tick $n$ using the following equations:

$$e_t := E(x_t) = W_{E_2} * (W_{E_2} * x_t + b_{E_1}) + b_{E_2} \tag{1}$$

$$c_d^n, h_d^n := \text{ConvLSTM}_d(e_t, h_{d-1}^n, c_d^{n-1}, h_d^{n-1}) \tag{2}$$

$$i_d^n := \tanh(W_{ii} * e_t + W_{ih_1} * h_{d-1}^n + W_{ih2} * h_d^{n-1} + b_i) \tag{3}$$

$$j_d^n := \sigma(W_{ji} * e_t + W_{jh_1} * h_{d-1}^n + W_{jh_2} * h_d^{n-1} + b_j) \tag{4}$$

$$f_d^n := \sigma(W_{fi} * e_t + W_{fh_1} * h_{d-1}^n + W_{fh_2} * h_d^{n-1} + b_f) \tag{5}$$

$$o_d^n := \tanh(W_{oi} * e_t + W_{oh_1} * h_{d-1}^n + W_{oh_2} * h_d^{n-1} + b_o) \tag{6}$$

$$c_d^n := f_d^n \odot c_d^{n-1} + i_d^n \odot j_d^n \tag{7}$$

$$h_d^n := o_d^n \odot \tanh(c_d^n) \tag{8}$$

Here $*$ denotes the convolution operator, and $\odot$ denotes point-wise multiplication. Note that $\theta_d = (W_{i\cdot}, W_{j\cdot}, W_{f\cdot}, W_{o\cdot}, b_i, b_j, b_f, b_o)_d$ parameterizes the computation of the $i, j, f, o$ gates. For the first ConvLSTM layer, the hidden state of the final ConvLSTM layer is used as the previous layer's hidden state.

A linear combination of the mean- and max-pooled ConvLSTM activations is injected into the next step, enabling quick communication across the receptive field, known as pool-and-inject. A boundary

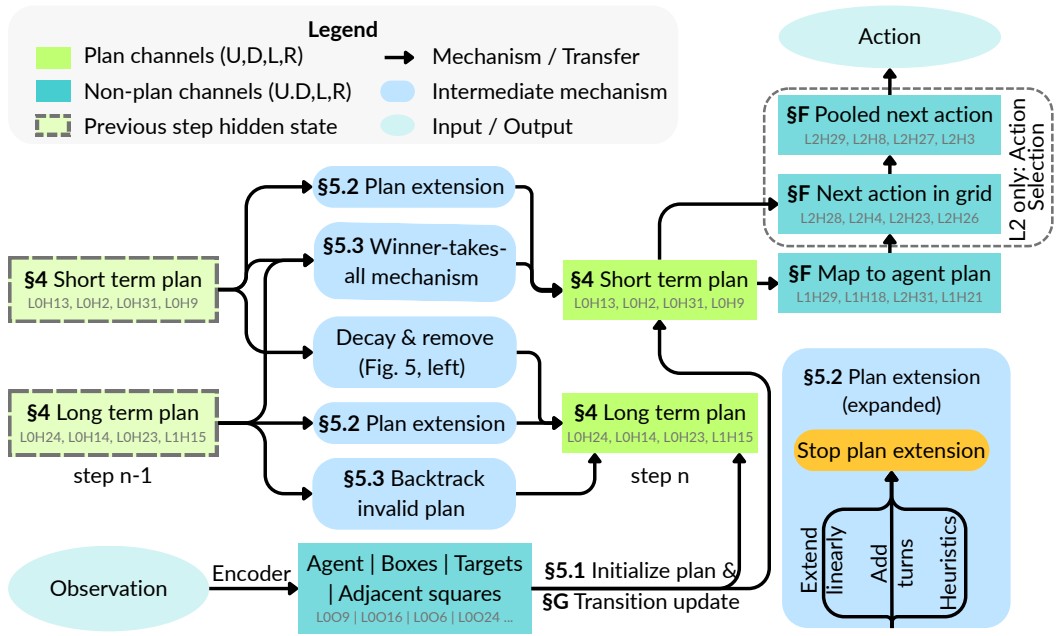

Figure 8: The planning algorithm learned by DRC$(3, 3)$. While the plan nodes are present and updated across all the layers, this circuit only shows the plan in the first layer's hidden state (L0HX) with a channel X for each direction up, down, left, and right. Mechanisms are annotated with the sub-section they are studied in Section 5.

feature channel with ones at the boundary of the input and zeros inside is also appended to the input. These are ignored in the above equations for brevity.

Finally, an MLP with 256 hidden units transforms the flattened ConvLSTM outputs $h_D^N$ into the policy (actor) and value function (critic) heads. In our setup, $D = N = 3$ and $C = 32$ matching Guez et al. [22]'s original hyperparameters. An illustration of the full architecture is shown in Figure 10.

**Encoder Simplification**    To interpret the weights of the encoder, we use the associativity of linear operations to combine the convolution operation of the encoder and the convolution kernels that map the encoder output $e_t$ to the hidden state $h_d^n$ into a single convolutional layer. For all $d > 0$ and $c \in \{i, j, f, o\}$, we define the combined kernel $W_{ce}^d$ and bias $b_{ce}^d$ as:

$$W_{ce}^d := W_{ii}^d * W_{e2} * W_{e1} \tag{9}$$

$$b_{ce}^d := W_{ii}^d * (b_{e2} + W_{e2} * b_{e1}) \tag{10}$$

$$W_{ii}^d * e_t = W_{ce}^d * x_t + b_{ce}^d \text{ (up to edge effects)} \tag{11}$$

## C   Network training details

The network was trained using the IMPALA V-trace actor-critic [18] reinforcement learning (RL) algorithm for $2 \cdot 10^9$ environment steps with Guez et al.'s Deep Repeating ConvLSTM (DRC) recurrent architecture consisting of three layers repeated three times per environment step, as shown in Figure 10.

The observations are $H \times W$ RGB images with height $H$ and width $W$. The agent, boxes, and targets are represented by the green ■, brown ■, and red ■ pixels respectively [54], as illustrated in Figure 9. The environment has -0.1 reward per step, +10 for solving a level, +1 for putting a box on a target and -1 for removing it.

**Dataset**    The network was trained on 900k levels from the unfiltered train set of the Boxoban dataset [21]. Boxoban separates levels into train, validation, and test sets with three difficulty levels:

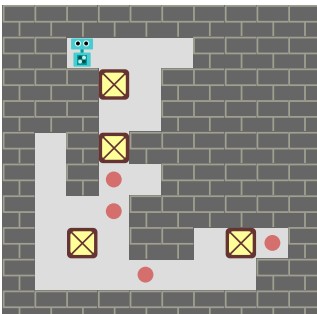 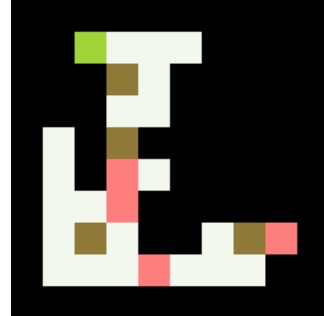

Figure 9: High resolution visualization of a Sokoban level along with the corresponding symbolic representation that the network observes. The agent, boxes, and targets are represented by the green ■, brown ■, and red ■ squares respectively.

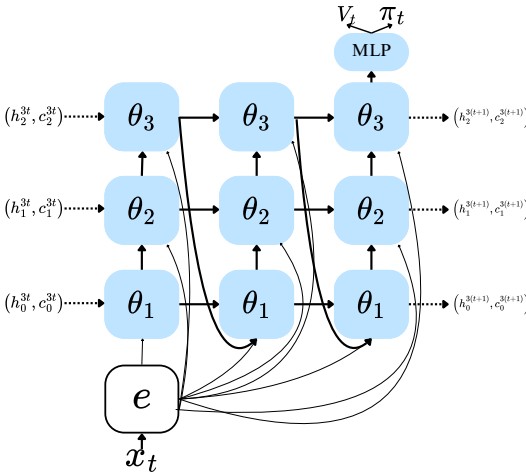

Figure 10: DRC$(3, 3)$architecture. Blocks parametrized by $\theta$ represent the ConvLSTM module shown in Figure 2. There are three layers of ConvLSTM modules with all the layers repeated applied three times before predicting the next action.

unfiltered, medium, and hard. The hard set is a single set with no splitting. Guez et al. [22] generated these sets by filtering levels unsolvable by progressively better-trained DRC networks. So easier sets occasionally contain difficult levels. Each level in Boxoban has 4 boxes in a grid size of $H = W = 10$. The $H \times W$ observations are normalized by dividing each pixel component by 255. The edge tiles in the levels from the dataset are always walls, so the playable area is $8 \times 8$. The player has four actions available to move in cardinal directions (Up, Down, Left, Right). The reward is -0.1 per step, +1 for placing a box on a target, -1 for removing it, and +10 for finishing the level by placing all of the boxes. In this paper, we evaluate the network on the validation-medium and hard sets of the Boxoban dataset. We also often evaluate the network on custom levels with different grid sizes and number of boxes to clearly demonstrate certain mechanisms in isolation.

**Action probe for evaluation on larger grid sizes**   The DRC$(3, 3)$network is trained on a fixed $H \times W$ grid size with the hidden state channels flattened to a $H \times W \times C$ tensor before passing it to the MLP layer for predicting action. Due to this limitation, the network cannot be directly evaluated on larger grid sizes. Taufeeque et al. [62] trained a probe using logitic regression with 135 parameters on the hidden state $h$ of the final ConvLSTM layer to predict the next action. They found that the probe can replace the 0.8M parameter MLP layer to predict the next action with a 77.9% accuracy. They used this probe to show that the algorithm learned by the DRC backbone generalizes to grid sizes 2-3 times larger in area than the training grid size of $10 \times 10$. We use these action probes to run the same network on larger grid sizes in this paper.

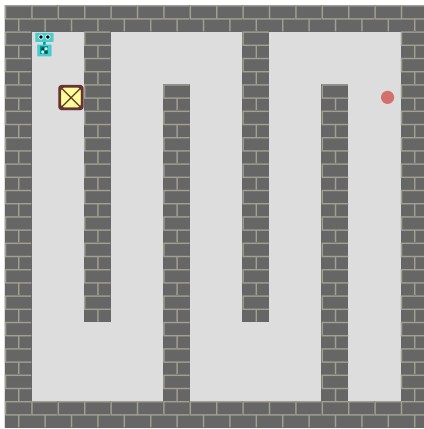

Figure 11: $16 \times 16$ zig-zag level that the original DRC$(3,3)$ network fails to solve. Steering $W_{ch1}^d$ and $W_{ch_2}^d$ by a factor of 1.2 solves this level and similar zig-zag levels for sizes upto $25 \times 25$.

Table 3: Comparison of network intervened with single-step cache across different channel groups. We report the percentage drop of solve rate compared to the original network (%) on medium-difficulty levels.

| Group | # Channels | Performance Drop |
|---|---|---|
| Non-planning | 37 | 10.5 |
| Planning | 59 | 57.6 |
| Random planning subset | 37 | 41.3 |

## D  Gate importance

We identify here the components that are important and others which can be ignored. We noticed that our analysis can be simplified by ignoring components like the previous cell-state $c$ and forget gate $f$ that don't have much effect. On mean-ablating the cell-state $c$ at the first tick $n = 0$ of every step for all the layers, we find that the network's performance drops by $21.28\% \pm 0.04\%$. The same ablation on the forget gate $f$ results in a drop of $2.66\% \pm 0.03\%$. On the other hand, the same ablation procedure on any of the other gates $i$, $j$, $o$, or the hidden state $h$ breaks the network and results in a drop of $100.00\%$ with no levels solved at all. This shows that the forget gate is not as important as other gates in regulating the information in the cell-state, and the information in the cell-state itself is not relevant for solving most levels. The only place we found the forget gates to be important is for accumulating the next-action in the GNA channels (Appendix F).

The mean-ablation experiment shows that the network computation from previous to the current step can be simplified to the following:

$$c_d^n \approx E[f_d^n] \odot E[c_d^{n-1}] + i_d^n \odot j_d^n = \mu + i_d^n \odot j_d^n \tag{12}$$
$$h_d^n = o_d^n \odot \tanh(c_d^n) \approx o_d^n \odot \tanh(\mu + i_d^n \odot j_d^n) \tag{13}$$

We therefore focus more on the $i, j, o$ gates and the hidden state $h$ in our analysis in this paper. Qualitatively, it also looks like the cell-state $c$ is very similar to the hidden state $h$. Note that the cell state $c$ not being much relevant doesn't imply that the network is not using information from previous hidden states, since most of the information from the previous hidden states $h_d^{n-1}$ flows through the $W_{ch_2}^d$ kernels.

## E  Label verification and offset computation

We see from Table 8 that most channels can be represented with some combination of features that can be derived from observation image (base feature) and future box or agent movements (future

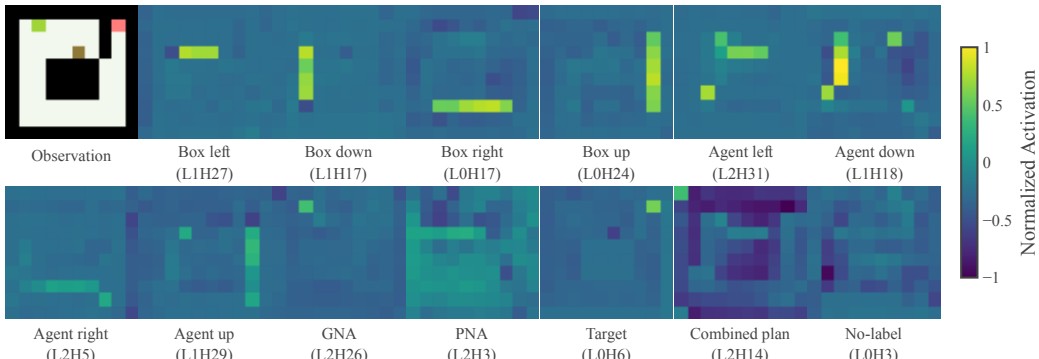

Figure 12: A toy observation with demonstrations of a single channel from every channel group in Table 7.

features). We compute the following 5 base features: agent, floor, boxes not on target, boxes on target, and empty targets. For future features, we get 3 features for each direction: box-movement, agent-movement, and a next-action feature that activates positively on all squares if that action is taken by the network at the current step. We perform a linear regression on the 5 base and 12 future features to predict the activations of each channel in the hidden state $h$.

**Offset computation** On visualizing the channels of the DRC$(3, 3)$network, we found that the channels are not aligned with the actual layout of the level. The channels are spatially-offset by a few squares in the cardinal directions. To automatically compute the offsets, we perform linear regression on the base and future features to predict the channel activation by shifting the features along $x, y \in \{-2, -1, 0, 1, 2\}$ and selecting the offset regression model with the lowest loss. The channels offsets are available in Table 4. We manually inspected all the channels and the offset and found that this approach accurately produces the correct offset for all the 96 channels in the network. All channel visualization in the paper are shown after correcting the offset.

**Correlation** The correlation between the predicted and actual activations of the channels is provided in the Tables 5 and 6. We find that box-movement, agent-movement, combined-plan, and target channels have a correlation of 66.4%, 50.8%, 48.0%, and 76.7%. As expected, the unlabeled channels do not align with our feature set and have the lowest correlation of 40.2%. Crucially, a baseline regression using only base features yielded correlations below 20% for all channels, confirming that the channels are indeed computing plans using future movement directions. These correlations should be treated as lower bounds, as this simple linear approach on the binary features cannot capture many activation dynamics like continuous development, representation of rejected alternative plans (Section 5.3), or the distinct encoding of short- vs. long-term plans.

# F Plan Representation to Action Policy

The plan formed by the box movement channels are transferred to the agent movement channels. For example, Figure 21b shows that the agent down movement channel L1H18 copies the box down movement channel L1H17 by shifting it one square up, corresponding to where the agent will push the box. The kernels also help in picking a single path if the box can go down through multiple paths.

Once the box-plan transfers to the agent-movement channels, these channels are involved in their own agent-path extension mechanism. Figure 21a show that the agent-movement channels have their own linear-plan-extension kernels. These channels also have stopping conditions that stop the plan-extension at the box squares and agent square. Thus, as a whole, the box-movement channels find box to target paths and the agent-movement channels copy those paths and also find agent to box paths.

Finally, the network needs to find the next action to take from the complete agent action plan represented in agent-movement channels. We find that the network dedicates separate channels that

extract the next agent action. We term these channels as the grid-next-action (GNA) channels (Table 7). There exists one GNA channel for each of the four action directions. A max-pooling operation on these channels transfers the high activation of an action to the entire grid of the corresponding agent action channel. We term these as the pooled-next-action (PNA) channels (Table 7). Lastly, the MLP layer aggregates the flattened neurons of the PNA channels to predict the next action. We verify that the PNA and GNA channels are completely responsible for predicting the next action by performing causal intervention that edits the activation of the channel based on our understanding to cause the agent to take a random action at any step in a level. Table 2 shows that both the PNA and GNA channels are highly accurate in modifying the next action. We now describe how the network extracts only the next agent move into the GNA channels.

The individual gates of the GNA channels copy activations of the agent-movement channels. Some gates perform subtraction of the agent and box movement channels to get agent-exclusive moves and the next agent box push. Figure 13 (top-right) shows one such example where the agent and box movement channels from layer 1 are subtracted resulting in an activation exclusively at the agent square. The GNA gates also receive positive activation on the agent square through L2H27 which detects agent at the first tick $n = 0$ of a step. Figure 13 shows that the $f$-gate of all GNA channels receives a positive contribution from the agent square. To counteract this, the agent-movement channels of one direction contribute negatively to the GNA channels of all other directions. All of this results in the agent square of the GNA channel of the next move activating strongly at the second tick $n = 1$.

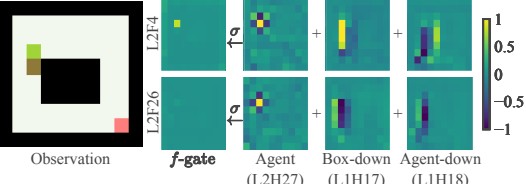

Figure 13: **Left:** Observation at step 3 where the agent moves down. **Right:** The GNA channels, which represent the direction that the agent will move in at the next step, predict the agent moving down primarily through $f$-gate. The box- and agent-down channels are offset and subtracted to get the action at the agent square. The checkered agent location pattern from L2H27 also helps in isolating the action on the agent square. The active $f$-gate square accumulates activation in the cell-state $c$ which after max-pooling and MLP layer decodes to the down action being performed.

Thus we have shown that the complete plan is filtered through the GNA channels to extract the next action which activates the PNA channel for the next action to be taken.

# G   State Transition Update

We have understood how the plan representation is formed and mapped to the next action to be taken. However, once an action is taken, the network needs to update the plan representation to reflect the new state of the world. We saw in Figure 3 that the plan representation is updated by deactivating the square that represented the last action in the plan. This allowed a different future action to be represented at the same square in the short-term channel which was earlier stored only in the long-term channel. We now show how a square is deactivated in the plan representation.

After an action is taken, the network receives the updated observation on the first tick $n = 0$ with the new agent or box positions. The combined $W_{ce}^d$ kernels for each layer that map to the path channels contain filters that detect only the agent, box, or target, often with the opposite sign of activation of the plan in the channel (Figure 22). Hence, when the observation updates with the agent in a new position, the agent kernels activates with the opposite sign of the plan activation that deletes the last move from the plan activation in the hidden state. The activation contributions in Figure 6 shows the negative contribution from the encoder kernels on the agent and the square to the left of the box. Therefore, the agent and the boxes moving through the level iteratively remove squares from the plan when they are executed with the plan-stopping mechanism ensuring that the plan doesn't over-extend beyond the new positions from the latest observation.

# H   Activation transfer mechanism from long to short term channels

Consider a scenario where two different actions, $A_1$ and $A_2$ ($A_1 \neq A_2$), are planned for the same location ("square") at different timesteps, $t_1$ and $t_2$, with $t_1 < t_2$. As illustrated in Section 5.3,

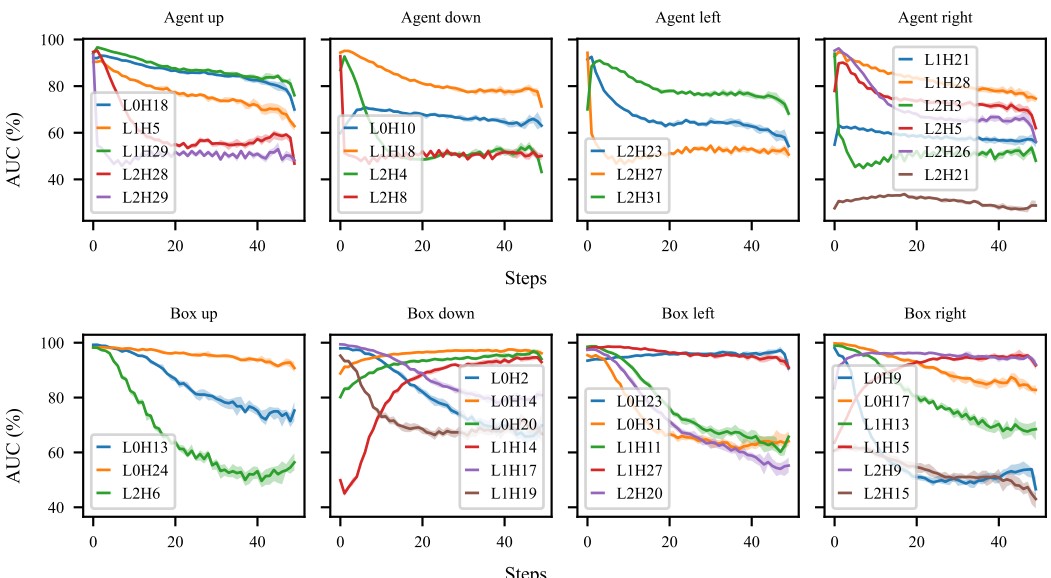

Figure 14: AUC scores of agent and boxes for all directions. The two channels in agent directions that quickly fall are the GNA/PNA channel which have a high AUC (100%) only for the next action. Short-term channel show a high AUC for predicting actions 10 steps in advance whereas the long-term channels show a high AUC for predicting agent's actions beyond 10 steps until the end of the episode.

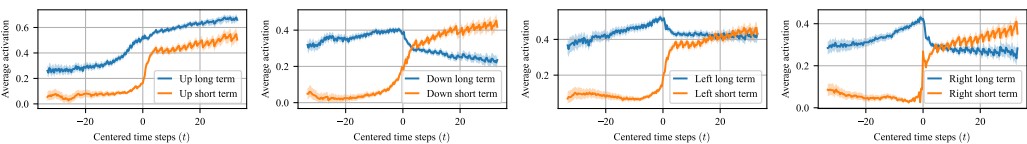

Figure 15: Activations of the long- and short-term channels for all directions when a different direction action takes place at $t = 0$. All direction except the up direction shows the long-term channel activations decreasing after the other action takes place at $t = 0$. The mechanism of this transfer of activation from long to short-term is shown in Figure 16.

Figure 3 (right) and further detailed in Figure 15, the later action ($A_2$ at $t_2$) is initially stored in the long-term channel for timesteps $t < t_1$. This information is transferred to the short-term channel only after the earlier action ($A_1$) is executed at $t = t_1$. We now describe the specific mechanism responsible for this transfer of activation from the long-term to the short-term channel.

In Figure 16, the activations transfer into L1H17 (short-term-down) from L0H14 (long-term-down) and L0H2 (short-term-down) channels when a right action is taken at $t = 0$ represented in L0H9 (short-term-right). The short-term-right channel L0H9 imposes a large negative contribution via the $j$-gate to inhibit L1H17, keeping it inactive even as the long-term-down channel tries to transfer a signal through the $j$ and $o$-gates for $t < 0$. Once the first move completes ($t = 0$), short-term-right is no longer active and so the inhibition ceases. The removal of the negative input allows the $j$-gate's activation to rise, enabling the long-term-down activation transfer through $o$-gate, making it the new active short-term action at the square. This demonstrates how long-term channels hold future plans, insulated from immediate execution conflicts by the winner takes all (WTA) mechanism (Section 5.3 and Figure 1) acting on short-term channels.

# I    Case study: Backtracking mechanism

Consider the level depicted in Figure 17 (a). The network begins by chaining forward from the box and backward from the target(Figure 17, b). Upon reaching the square marked D1, the plan can continue upwards or turn left. Here, the turn and linear plan-extension kernels activate the box-right and the box-down channels respectively. However, box-down activation is much higher because the

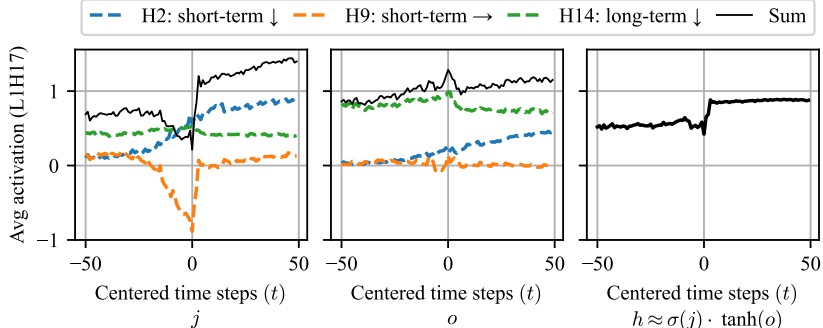

Figure 16: Transfer mechanism from long to short-term channel shown through contributions into the gates of the short-term-down (L1H17) channel averaged across squares where a right box-push happens at $t = 0$ and down box-push later on. The long-term-down channel L0H14 contributes to the $o$-gate at all steps $t$. However, L0H9 (short-term-right) activates negatively in the sigmoid $j$-gate, thus deactivating L1H17. As the right move gets played at $t = 0$, L0H9's negative contribution vanishes, enabling the transfer of L0H14 and L0H2 into L1H17.

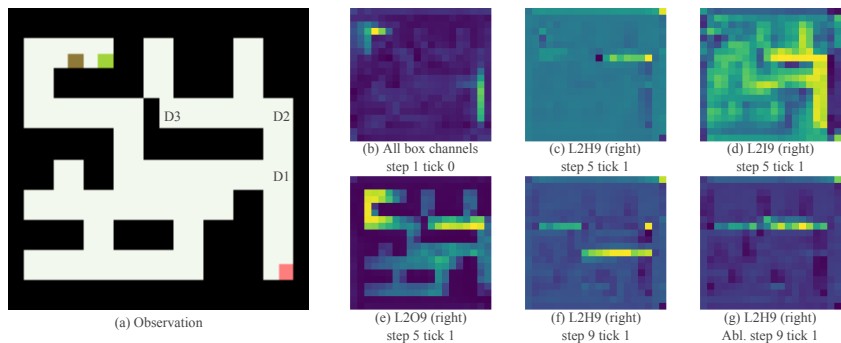

Figure 17: **(a)** $20 \times 20$ level we term as the "backtrack level" with key decision nodes D1-D3 for backward chaining. **(b)** The sum of box-movement channels at step 1 tick 0 indicates forward (from box) and backward (from target) chaining. **(c-e)** Activation of the box-right channel L2H9 involved in backward chaining at step 5 tick 1. Backward chaining moved up from D1 to D2 and then hitting a wall at D3, which initiates backtracking towards D2 through negative plan extension. The negative wall activation comes from the $o$-gate of L2H9. **(f)** Successful pathfinding at T28 after backtracking redirected the search. **(g)** Ablation: Forcing positive activation at D3 (by setting it to its absolute value) prevents backtracking, hindering correct solution finding (L2H9 Abl., T28).

In particular, the forced positive ablation at D3 results in an incorrect plan (g) which seemingly goes right all the way through the wall, as opposed to the correct plan (f) which goes right on a valid path.

weights of the linear extension kernels are much larger than the turn kernels (as seen in Figure 1). Due to this, the winner-takes-all mechanism leads to the search continuing upwards in the box-down channel. Upon hitting a wall at D2, the chain turns right along the 'box-right channel' (L2H9) and continues until it collides with another wall at D3. (Figure 17, c).

This triggers backtracking. While both $i$-gate and $o$-gate activations contribute to plan extension, the $o$-gate also activates strongly *negatively* on wall squares like D3 (Figure 17d, e). This leads to a dominant negative activation in the 'box-right' channel, which then propagates backward along the explored path (from D3 towards D2) via the forward plan-extension kernels of L2H9.

This weakens the dominant 'box-down' activation at D1, allowing the alternative 'box-right' path from D1 to activate. The search then proceeds along this new route, allowing the backward chain to connect with the forward chain, resulting in the correct solution (Figure 17, f).

To verify this mechanism, we performed an intevention by forcing the activation at the wall squares near D3 to be positive (by taking their absolute values). This blocked backtracking, and the network

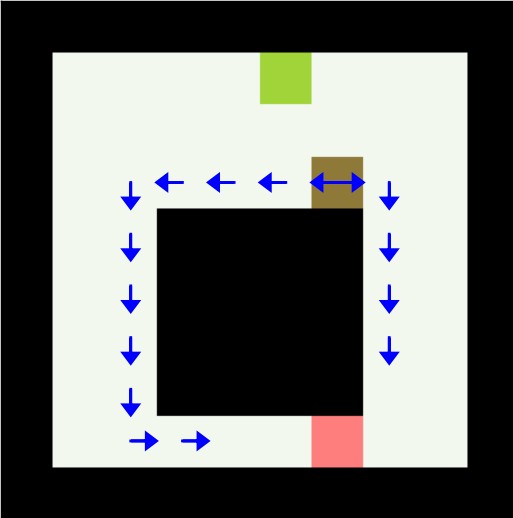

Figure 18: A level with two paths, one longer than the other. We initialize the starting hidden state with the two paths shown such that they both have two squares left to reach the target. We find that the expands both paths and picks the left (longer) path through the winner-takes-all mechanism since it reaches there with higher activation through linear-plan-extension.

incorrectly attempted to connect the chains through the wall (Figure 17, g). This confirms that negative activation generated at obstacles is the key driver for backtracking, and is what allows the network to discard failed paths and explore alternatives. We quantitatively test this claim further by performing the same intervention on transitions from 512 levels where a plan's activation is reduced by more than half in a single step which was preceded by a neighboring square having negative activation in the path channel. We define the intervention successful if forcing the negative square to an absolute value doesn't reduce the activation of the adjacent plan square. The intervention results in a success rate with 95% confidence intervals of $85.1\% \pm 5.0\%$ and $48.9\% \pm 3.3\%$ for long- and short-term channels, respectively. This checks out with the fact that long-term channels represent plans not in the immediate future which would get backtracked through negative path activations. On the other hand, negative activations in the short-term channels are also useful during the winner-takes-all (WTA) mechanism and deadlock prevention heuristics. Filtering such activations for short-term channels from the intervention dataset would improve the numbers.

## J  Case study: making the network take the longer path

The network usually computes the shortest paths from a box to a target by forward (from box) and backward (from target) chaining linear segments until they connect at some square as illustrated in Figure 1. As soon as a valid plan is found for a box along one direction, the winner-takes-all mechanism stabilizes that plan through its stronger activations and deletes any other plans being searched for the box. From this observation, we hypothesize that the network values finding valid plans in least number of steps than picking the shorter one. We verify this value preference of the network by testing the network with on the level shown in Figure 18 with the starting state initialized with the two paths shown. The left path (length=13) is longer than the right path (length=7) for reaching from the box to target. Both paths are initialized in the starting hidden state to have two arrow left to complete the path. We find that in this case, both the paths reach the target, but the left one is stronger due to linear plan extension kernels reaching with higher activation. This makes the network pick the left path and prune out the shorter right path. If we modify the starting state such that left and right paths have 3 and 2 square left to the reach the target, then the right path wins and the left path is pruned out. This confirms that the network's true value in this case is to pick a valid plan closer to target than to pick a shorter plan. However, since convolution moves plan one square per operation, the network usually seems to have the value of picking the shorter plan.

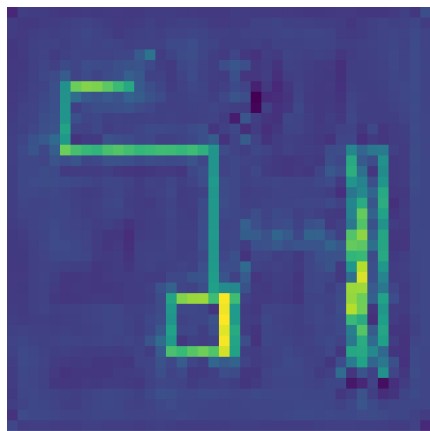

Figure 19: Sum of activations of box-movement channels on the $40 \times 40$ backtrack level with the network weights $W_{ch1}^d$ and $W_{ch_2}^d$ steered by a factor of 1.4. The planning representation gets stuck in the loop shown, unable to backtrack and explore other paths. The activations of other squares become chaotic, changing rapidly and randomly on each step.

## K  Unsuccessful methods

Section 3 describes the methods we found useful to understand the learned algorithm of the DRC$(3, 3)$ network. We also tried the following popular interpretability methods but found them to not work well for our network: Network Pruning, Automated Circuit Discovery, explicitly coding the causal abstract graph [6], and Sparse Autoencoders.

## L  Channel Redundancy

We see from Table 7 that the network represents many channels per box-movement and agent-movement direction. We find at least two reasons for why this redundancy is useful.

First, it facilitates faster spatial propagation of the plan. Since the network uses $3 \times 3$ kernels in the ConvLSTM block, information can only move 1 square in each direction per convolution operation. By using redundant channels across multiple layers, the network can effectively move plan information several squares within a single time step's forward pass (one square per relevant layer). Evidence for this rapid propagation is visible in Figure 17(b), where plan activations extend 7-10 squares from from the target and the box within the first four steps on a $20 \times 20$ level.

Second, the network dedicates separate channels to represent the plan at different time horizons. We identified distinct short-term (approximately 0-10 steps ahead) and long-term (approximately 10-50 steps ahead) channels within the box and agent-movement categories.

This allows the network to handle scenarios requiring the same location to be traversed at different future times. For example, if a box must pass through the same square at time $t_1$ and later at time $t_2$, the network can use the short-term channel to represent the first push at $t_1$ and the long-term channel to represent the second push at $t_2$. Figure 3 (right) illustrates this concept, showing activation transferring from a long-term to a short-term box-down-movement channel once the earlier action at that square is taken by the agent.

## M  Weight steering fixes failure on larger levels

Previous work [62] showed that, although the DRC$(3, 3)$ network can solve much bigger levels than $10 \times 10$ grid size on which it was trained, it is easy to contruct simple and natural adversarial examples which the network fails to solve. For example, the $n \times n$ zig-zag level in Figure 11 that can be scaled arbitrarily by adding more alleys and making them longer, is only solved for $n \leq 15$ and fails on all $n > 15$. The big level shown in Figure 17 (a) is solved by the network on the $20 \times 20$ grid size but fails on $30 \times 30$ or $40 \times 40$ grid size.

Figure 20 (a) visualizes the sum of activations of the box-movement channels on a $40 \times 40$ variant of the backtrack level in which we see the reason why larger levels fail: the channel activations decay as the plan gets extended further and further. This makes sense as the network only saw $10 \times 10$ levels during training and hence the kernel weights were learned to only be strong enough to solve levels where targets and boxes are not too far apart. We find that multiplying the weights of $W_{ch1}^d$ and $W_{ch_2}^d$, the kernels that update and maintain the hidden state, by a factor of $1.2$ helps the network extend the plan further. This weight steering procedure is able to solve the zig-zag levels for sizes up to $n = 25$ and the backtrack level for sizes up to $40 \times 40$. Figure 20 (b, c) show that upon weight steering, the box-movement channels are able to maintain their activations for longer, enabling the network to solve the level. However, for much larger levels, weightsteered networks also fall into the same trap of decaying activations, failing to extend the plan. Further weight steering with a larger factor can help but we find that it can become brittle, as the planning representation gets stuck in wrong paths, unable to backtrack, with the activations becoming chaotic (Figure 19). We also tried other weight steering approaches such as multiplying all the weights of the network by a factor or a subset such as the kernels of path channels, but find that they do not work as well as the weight steering of $W_{ch1}^d$ and $W_{ch_2}^d$.

# N    Limitations

Our paper has several limitations. First, we only reverse-engineer one $\text{DRC}(3,3)$ network in this paper. We are fairly confident that our results generalize to any $\text{DRC}(3,3)$ network trained on Sokoban using model-free reinforcement learning, but can't prove it. The network having similar performance and capabilities such as utilizing extra test-time compute across multiple papers who trained it independently suggests that the learned algorithm is a pretty stable minima [22, 62, 3].

Second, we only reverse-engineer DRC and no other networks. It is possible that the inductive biases of other networks such as transformer, Conv-ResNet, or 1D-LSTM may end up learning an algorithm that is different from what we found. Our results are also only on Sokoban and it is possible that the learned algorithm for other game-playing network looks very different from the one learned for Sokoban.

We also do not fully reverse-engineer the network. We have observed the following behaviors that cannot be explained yet with our current understanding of the learned algorithm:

- Agent sometimes does a bit of box 1, then box 2, then back to box 1, to minimize distance. Our explanation doesn't account for how and when the network switches between boxes.
- Sometimes the heuristics inexplicably choose where to go based on seemingly irrelevant things. Slightly changing the shape or an obstacle or moving the agent's position by 1 can sometimes change which plan gets chosen, in a manner that doesn't correspond to optimal plan.

# O    Societal Impact

This research into interpretability can make models more transparent, which helps in making models predictable, easier to debug and ensure they conform to specifications.

Specifically, we analyze a model organism which is planning. We hope that this will catalyze further research on identifying, evaluating and understanding what *goal* a model has. We hope that directly identifying a model's goal lets us monitor for and correct goal misgeneralization [14].

Table 7: Grouped channels and their descriptions. * indicates long-term channels.

| Group | Description | Channels |
|---|---|---|
| Box up | Activates on squares from where a box would be pushed up | L0H13, L0H24*, L2H6 |

Continued on next page

Table 7: Grouped channels and their descriptions. * indicates long-term channels.

| Group | Description | Channels |
|---|---|---|
| Box down | Activates on squares from where a box would be pushed down | L0H2, L0H14*, L0H20*, L1H14*, L1H17, L1H19 |
| Box left | Activates on squares from where a box would be pushed left | L0H23*, L0H31, L1H11, L1H27, L2H20 |
| Box right | Activates on squares from where a box would be pushed right | L0H9, L0H17, L1H13, L1H15*, L2H9*, L2H15 |
| Agent up | Activates on squares from where an agent would move up | L0H18, L1H5, L1H29, L2H28, L2H29 |
| Agent down | Activates on squares from where an agent would move down | L0H10, L1H18, L2H4, L2H8 |
| Agent left | Activates on squares from where an agent would move left | L2H23, L2H27, L2H31 |
| Agent right | Activates on squares from where an agent would move right | L1H21, L1H28, L2H3, L2H5, L2H21*, L2H26 |
| Combined Plan | Channels that combine plan information from multiple directions | L0H15, L0H16, L0H28, L0H30, L1H0, L1H4, L1H8, L1H9, L1H20, L1H25, L2H0, L2H1, L2H13, L2H14, L2H17, L2H18, L0H7, L0H1, L0H21, L1H2, L1H23, L2H11, L2H22, L2H24, L2H25, L2H12, L2H16, L0H19, L2H30 |
| Entity | Highly activate on target tiles. Some also activate on agent or box tiles | L0H6, L0H26, L1H6, L1H10, L1H22, L1H31, L2H2, L2H7 |
| No label | Uninterpreted channels. These channels do not have a clear meaning but they are also not very useful | L0H0, L0H3, L0H4, L0H5, L0H8, L0H22, L0H25, L0H27, L0H29, L1H1, L1H3, L1H12, L1H16, L1H26, L1H30, L2H10, L2H19, L0H11, L0H12, L1H7, L1H24 |
| Grid-Next-Action (GNA) | Channels that activate on squares that the agent will move in the next few moves. One separate channel for each direction | L2H28 (up), L2H4 (down), L2H23 (left), L2H26 (right) |
| Pooled-Next-Action (PNA) | A channel for each action that activates highly across all squares at the last tick ($n = 2$) to predict the action | L2H29 (up), L2H8 (down), L2H27 (left), L2H3 (right) |

Table 8: Informal description of all channels

| Channel | Long-term | Description |
|---|---|---|
| L0H0 | No | some box-left-moves? |
| L0H1 | No | box-to-target-lines which light up when agent comes close to the box. |
| L0H2 | No | H/-C/-I/J/-O: +future box down moves [1sq left] |
| L0H5 | No | [1sq left] |
| L0H6 | No | H/-C: +target -box -agent . F: +agent +agent future pos. I: +agent. O: -agent future pos. J: +target -agent[same sq] |
| L0H7 | No | (0.37 corr across i,j,f,o). |
| L0H9 | No | -H/-C/-O/I/J/F: +agent +future box right moves -box. -H/J/F: +agent-near-future-down-moves [on sq] |
| L0H10 | No | H: -agent-exclusive-down-moves [1sq left,down]. Positively activates on agent-exclusive-up-moves. |
| L0H11 | No | H: CO. O: box-right moves C/I: -box future pos [1sq up (left-right noisy)] |

Continued on next page

Table 8: Informal description of all channels

| Channel | Long-term | Description |
|---|---|---|
| L0H12 | No | H: very very faint horizontal moves (could be long-term?). I/O: future box horizontal moves (left/right). [on sq] |
| L0H13 | No | H/C/I/J/O: +future box up moves [1sq up] |
| L0H14 | Yes | H/-I/O/C/H: -future-box-down-moves. Is more future-looking than other channels in this group. Box down moves fade away as other channels also start representing them. Sometimes also activates on -agent-right-moves [on sq] |
| L0H15 | No | H/I/J/-F/-O: +box-future-moves. More specifically, +box-down-moves +box-left-moves. searchy (positive field around target). (0.42 corr across i,j,f,o). |
| L0H16 | No | H +box-right-moves (not all). High negative square when agent has to perform DRU actions. [1sq up,left] |
| L0H17 | No | H/I/J/F/O: +box-future-right moves. O: +agent [1sq up] |
| L0H18 | No | H: -agent-exclusive-up-moves |
| L0H20 | Yes | H: box down moves. Upper right corner positively activates (0.47 start -> 0.6 in a few steps -> 0.7 very later on). I: -box down moves. O: +box down moves -box horizontal moves. [1sq up] |
| L0H21 | No | -box-left-moves. +up-box-moves |
| L0H23 | Yes | H/C/I/J/O: box future left moves [1sq up,left] |
| L0H24 | Yes | H/C/I/J/O: -future box up moves. long-term because it doesn't fade away after short-term also starts firing [1sq up,left] |
| L0H26 | No | H: -agent . I/C/-O: all agent future positions. J/F: agent + target + BRwalls, [1sq up] |
| L0H28 | No | H/C/I/J/F/-O: -future box down moves (follower?) [on sq]. Also represents agent up,right,left directions (but not down). |
| L0H30 | No | H/I: future positions (0.47 corr across i,j,f,o). |
| L1H0 | No | H: -agent -agent near-future-(d/l/r)-moves + box-future-pos [on sq] |
| L1H2 | No | -box-left-moves |
| L1H4 | No | +box-left moves -box-right moves [1sq up]. |
| L1H5 | No | H: +agent-exclusive-future-up moves [2sq up, 1sq left] |
| L1H6 | No | J: player (with fainted target) |
| L1H7 | No | H: - some left box moves or right box moves (ones that end at a target)? Sometimes down moves? (unclear) |
| L1H8 | No | box-near-future-down-moves(-0.4),agent-down-moves(+0.3),box-near-future-up-moves(+0.25) [on sq] |
| L1H9 | Yes | O/I/H: future pos (mostly down?) (seems to have alternate paths as well. Ablation results in sligthly longer sols on some levels). Fence walls monotonically increase in activation across steps (tracking time). [on sq] |
| L1H10 | No | J/H/C: -box + target +agent future pos. (neglible in H) O,-I: +agent +box -agent future pos [1sq up] (very important feature – 18/20 levels changed after ablation) |
| L1H11 | No | -box-left-moves (-0.6). |
| L1H13 | No | H: box-right-moves(+0.75),agent-future-pos(+0.02) [1sq left] |
| L1H14 | Yes | H: longer-term down moves? [1sq up] |
| L1H15 | Yes | H/-O: box-right-moves-that-end-on-target (with high activations towards target). Activates highly when box is on the left side of target [on sq]. |
| L1H17 | No | H/C/I/-J/-F/O: -box-future down moves [on sq] |
| L1H18 | No | H/-O: +agent future down moves (stores alternate down moves as well?) [on sq] |
| L1H19 | No | H/-F/-J: -box-down-moves (follower?) [1sq up] |
| L1H20 | No | +near-future-all-box-moves [1sq up]. |
| L1H21 | No | H: agent-right-moves(-0.5) (includes box-right-pushes as well) |
| L1H22 | No | -target |
| L1H23 | No | -box-left-moves. |

Table 8: Informal description of all channels

| Channel | Long-term | Description |
|---|---|---|
| L1H24 | No | H: -box -agent-future-pos -agent, [1sq left] |
| L1H25 | No | all-possible-paths-leading-to-targets(-0.4),agent-near-future-pos(-0.07),walls-and-out-of-plan-sqs(+0.1),boxes(+0.6).     H: +box -agent -empty -agent-future-pos | O/-C: -agent +future sqs (probably doing search in init steps) | I: box + agent + walls | F: -agent future pos | J: +box +wall -agent near-future pos [1sq up,left] |
| L1H27 | No | H: box future left moves [1sq left] |
| L1H28 | No | some-agent-exclusive-right-moves(+0.3),box-up-moves-sometimes-unclear(-0.1) |
| L1H29 | No | agent-near-future-up-moves(+0.5) (~5-10steps, includes box-up-pushes as well). I: future up moves (~almost all moves) + agent sq [1sq up] |
| L1H31 | No | H: squares above and below target (mainly above) [1sq left & maybe up] |
| L2H0 | No | -box-all-moves. |
| L2H1 | No | H/O: future-down/right-sqs [1sq up] |
| L2H2 | No | H: high activation when agent is below a box on target and similar positions. walls at the bottom also activate negatively in those positions. |
| L2H3 | No | H: +right action (PNA) + future box -down -right moves + future box +left moves |
| L2H4 | No | O: +near-future agent down moves (GNA). I: +agent/box future pos [1sq left] |
| L2H5 | No | H/C/I/J: +agent-future-right-incoming-sqs, O: agent-future-sqs [1sq up, left] |
| L2H6 | No | H: +box-up-moves (~5-10 steps). -agent-up-moves. next-target (not always) [1q left] |
| L2H7 | No | +unsolved box/target |
| L2H8 | No | down action (PNA). |
| L2H9 | Yes | H/C/I/J/O: +future box right moves [1sq up] |
| L2H11 | No | -box-left-moves(-0.15),-box-right-moves(-0.05) |
| L2H13 | No | H: +box-future-left -box-long-term-future-right(fades 5-10moves before taking right moves) moves. Sometimes blurry future box up/down moves [1sq up] |
| L2H14 | No | H: all-other-sqs(-0.4) agent-future-pos(+0.01) O: -agent-future-pos. I: +box-future-pos |
| L2H15 | No | -box-right-moves [1sq up,left] |
| L2H17 | No | H/C: target(+0.75) box-future-pos(-0.3). O: target. J: +target -agent +agent future pos. I/F: target. [1sq up] |
| L2H18 | No | box-down/left-moves(-0.2). Very noisy/unclear at the start and converges later than other box-down channels. |
| L2H19 | No | H: future agent down/right/left sqs (unclear) [1sq up] |
| L2H20 | No | H: -box future left moves [1sq left] |
| L2H21 | Yes | H: -far-future-agent-right-moves. Negatively contributes to L2H26 to remove far-future-sqs. Also represents -agent/box-down-moves. [1sq up] |
| L2H22 | No | H: box-right-moves(+0.3),box-down-moves(0.15). O future sqs??? |
| L2H23 | No | H: future left moves (does O store alternate left moves?) (GNA). [1sq left] |
| L2H24 | No | box-right/up-moves (long-term) |
| L2H25 | No | unclear but (8, 9) square tracks value or timesteps (it is a constant negative in the 1st half episode and steadily increases in the 2nd half)? |
| L2H26 | No | H/O: near-future right moves (GNA). [on sq] |
| L2H27 | No | left action (PNA). T0: negative agent sq with positive sqs up/left. |
| L2H28 | No | near-future up moves (GNA). O: future up moves (not perfectly though) [1sq up] |
| L2H29 | No | Max-pooled Up action channel (PNA). |
| L2H31 | No | some +agent-left-moves (includes box-left-pushes). |

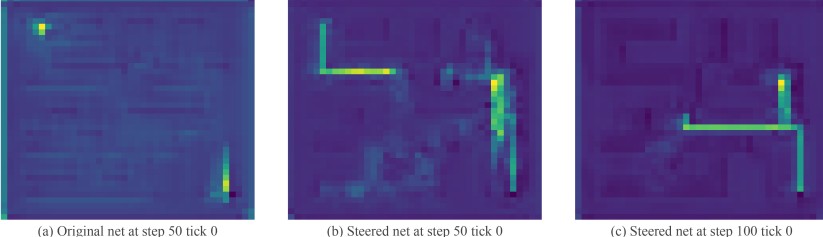

(a) Original net at step 50 tick 0    (b) Steered net at step 50 tick 0    (c) Steered net at step 100 tick 0

Figure 20: The sum of activations of the box-movement channels on a $40 \times 40$ variant of the backtrack level from Figure 17 for **(a)** the original network at step 50, and the weight-steered network at **(b)** step 50 and **(c)** step 100 when the agent reaches halfway through. The original network fails to solve the level as the plan decays and cannot be extended beyond $10 - 15$ squares. Upon weight steering, the plan activations travel farther without decaying thus solving the level.

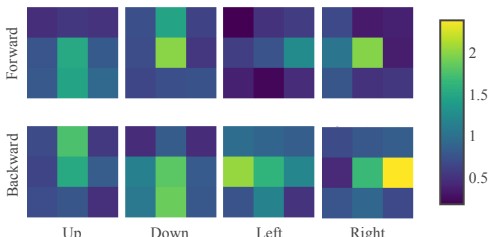

(a) Forward and backward plan extension kernels averaged over agent-movement channels. Agent-movement channels also extend the agent moves forward and backward similar to the box-plan extension.

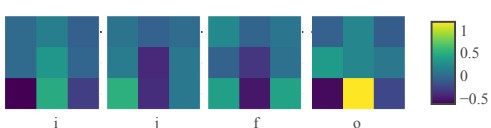

(b) The kernels that map L1H17 (box-down) to L1H18 (agent-down) by shifting the activation one square up. L1H17 activates negatively, therefore the $j$ and $f$ kernels are negative since they use the sigmoid activation function. The $i$ and $o$ kernels are positive which results in negatively activating $i$ and $o$-gates, which after multiplication results in L1H18 activating positively. The opposite signed weights on the lower-corner squares of the kernel help in picking a single path out of multiple parallel paths.

Figure 21: Plan extension and box path to agent path kernels.

Table 4: Activation offset along (row, column) in the grid for each layer and channel

|  | Layer 0 | Layer 1 | Layer 2 |
| --- | --- | --- | --- |
| Channel 0 | (1, 0) | (0, 0) | (-1, 0) |
| Channel 1 | (0, 0) | (-1, -1) | (-1, -1) |
| Channel 2 | (0, -1) | (-1, 0) | (0, 0) |
| Channel 3 | (0, 0) | (-1, 0) | (0, 0) |
| Channel 4 | (-1, -1) | (-1, -1) | (0, -1) |
| Channel 5 | (0, -1) | (-2, -1) | (-1, 0) |
| Channel 6 | (0, 0) | (-1, -1) | (-1, -1) |
| Channel 7 | (-1, 0) | (-1, 0) | (0, 0) |
| Channel 8 | (0, -1) | (0, 0) | (-1, 0) |
| Channel 9 | (0, 0) | (0, 0) | (-1, 0) |
| Channel 10 | (-1, -1) | (-1, 0) | (-1, 0) |
| Channel 11 | (-1, 0) | (0, -1) | (0, -1) |
| Channel 12 | (0, -1) | (0, -1) | (0, -1) |
| Channel 13 | (-1, 0) | (-1, 0) | (-1, 0) |
| Channel 14 | (0, 0) | (0, -1) | (-1, -1) |
| Channel 15 | (0, 0) | (0, 0) | (-1, -1) |
| Channel 16 | (-1, -1) | (0, 0) | (-1, -1) |
| Channel 17 | (-1, 0) | (0, 0) | (-1, 0) |
| Channel 18 | (-1, 0) | (0, 0) | (-1, 0) |
| Channel 19 | (-1, -1) | (-1, 0) | (-1, -1) |
| Channel 20 | (-1, 0) | (0, -1) | (0, -1) |
| Channel 21 | (-1, 0) | (-1, 0) | (0, 0) |
| Channel 22 | (0, 0) | (0, 0) | (-1, 0) |
| Channel 23 | (-1, -1) | (-1, 0) | (0, -1) |
| Channel 24 | (-1, -1) | (0, -1) | (-1, 0) |
| Channel 25 | (-1, 0) | (-1, -1) | (-1, -1) |
| Channel 26 | (-1, 0) | (0, -1) | (0, 0) |
| Channel 27 | (-1, -1) | (-1, -1) | (0, 0) |
| Channel 28 | (0, 0) | (0, 0) | (-1, 0) |
| Channel 29 | (0, 0) | (-1, 0) | (0, -1) |
| Channel 30 | (-1, 0) | (0, 0) | (-1, -1) |
| Channel 31 | (-1, -1) | (0, -1) | (0, -1) |

Table 5: Correlation of linear regression model's predictions with the original activations for each channel.

|  | Layer 0 | Layer 1 | Layer 2 |
|---|---|---|---|
| Channel 0 | 33.15 | 79.48 | 70.03 |
| Channel 1 | 50.76 | 48.77 | 38.37 |
| Channel 2 | 73.15 | 28.90 | 39.17 |
| Channel 3 | 31.73 | 68.30 | 55.72 |
| Channel 4 | 45.06 | 50.10 | 45.64 |
| Channel 5 | 63.91 | 42.95 | 55.27 |
| Channel 6 | 96.57 | 87.47 | 53.90 |
| Channel 7 | 51.98 | 36.88 | 95.63 |
| Channel 8 | 46.64 | 41.58 | 55.04 |
| Channel 9 | 70.52 | 37.44 | 71.47 |
| Channel 10 | 37.68 | 99.01 | 53.91 |
| Channel 11 | 52.09 | 61.55 | 42.26 |
| Channel 12 | 41.54 | 43.86 | 27.19 |
| Channel 13 | 79.54 | 73.35 | 54.40 |
| Channel 14 | 72.17 | 48.12 | 56.54 |
| Channel 15 | 44.09 | 65.72 | 36.37 |
| Channel 16 | 63.49 | 26.56 | 38.24 |
| Channel 17 | 76.70 | 73.94 | 94.78 |
| Channel 18 | 61.51 | 66.11 | 34.18 |
| Channel 19 | 46.05 | 44.01 | 33.48 |
| Channel 20 | 65.00 | 58.94 | 64.92 |
| Channel 21 | 22.05 | 57.36 | 60.21 |
| Channel 22 | 26.51 | 63.73 | 24.32 |
| Channel 23 | 74.39 | 31.32 | 44.64 |
| Channel 24 | 83.64 | 58.56 | 59.94 |
| Channel 25 | 17.10 | 82.43 | 28.29 |
| Channel 26 | 75.48 | 44.26 | 45.17 |
| Channel 27 | 9.24 | 85.84 | 49.92 |
| Channel 28 | 46.87 | 42.65 | 15.38 |
| Channel 29 | 28.60 | 64.77 | 54.68 |
| Channel 30 | 47.70 | 35.00 | 40.15 |
| Channel 31 | 53.12 | 56.81 | 59.63 |

Table 6: Correlation of linear regression model's predictions with the original activations averaged over channels for each group. Includes correlation using only base features for comparison. The (all dir) group is the average of the four directions. NGA and PNA are included in the Agent groups.

| Group | Correlation | Base correlation |
|---|---|---|
| Box up | 72.36 | 21.01 |
| Box down | 62.73 | 13.93 |
| Box left | 67.96 | 21.10 |
| Box right | 65.69 | 27.40 |
| Box (all dir) | 66.37 | 20.83 |
| Agent up | 47.86 | 12.69 |
| Agent down | 51.12 | 15.85 |
| Agent left | 51.40 | 7.85 |
| Agent right | 52.73 | 14.92 |
| Agent (all dir) | 50.80 | 13.33 |
| Combined path | 48.00 | 23.35 |
| Entity | 76.73 | 70.66 |
| No label | 40.25 | 15.53 |

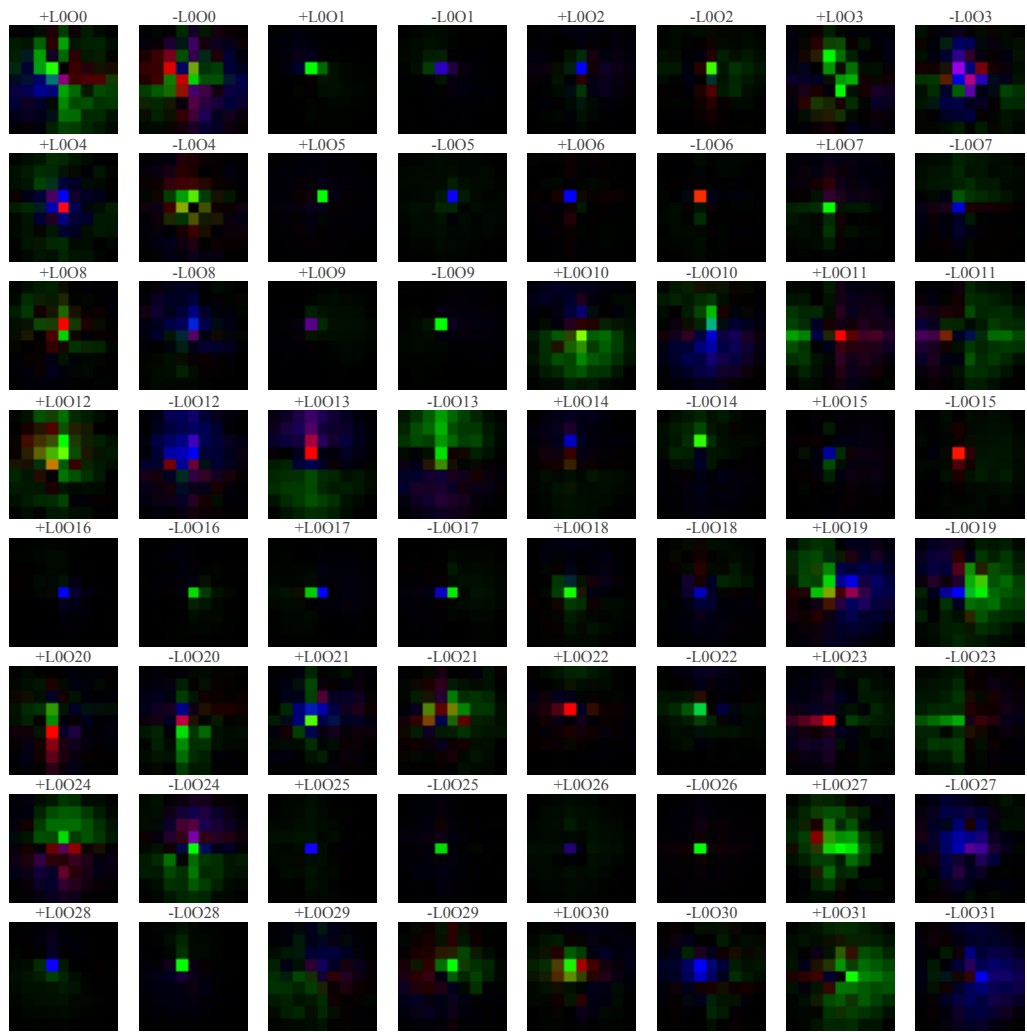

Figure 22: $9 \times 9$ combined convolutional filters $W_{oe}^0$ that map the RGB observation image to the $O$ gate in layer 0. The positive and negative components of each channel filters are separated visualized by computing $max(0, W_{oe}^0)$ and $max(0, -W_{oe}^0)$ respectively. The green, red, and brown colors in the filters detect the agent, target, and box squares respectively. The blue component is high only in empty tiles, so the blue color can detect empty tiles. We find that many filters are responsible for detecting the agent and the target like L0O5 and L0O6. A use case of such agent and box detecting filters in the encoder is shown in Figure 6. Many filters detect whether the agent or the target are some squares away in a particular direction like L0O20 and L0O23. Filters for other layers and gates can be visualized using our codebase.

Table 9: Solve rate (%) of different models without and with 6 thinking steps on held out sets of varying difficulty.

| Model | No Thinking | | | Thinking | | |
|---|---|---|---|---|---|---|
| | Hard | Med | Unfil | Hard | Med | Unfil |
| DRC(3, 3) | 42.8 | 76.6 | 99.3 | 49.7 | 81.3 | 99.7 |
| DRC(1, 1) | 7.8 | 28.1 | 89.4 | 9.8 | 33.9 | 92.6 |
| ResNet | 26.2 | 59.4 | 97.9 | - | - | - |

