# OpenReview forum: "Interpreting learned search: finding a transition model and value function in an RNN that plays Sokoban"
_NeurIPS.cc/2025/Conference — Submitted to NeurIPS 2025_

### Official Review · Reviewer_8R7r · 2025-06-05

**Clarity:** 2
**Significance:** 2
**Originality:** 3
**Rating:** 4
**Confidence:** 5

**Summary:**

This work is a follow up to [Guez 2019], that showed that Neural networks can perform planning, whihc was itself a follow up to the work on Value Iteration Networks [Tamar 2016].
A convolutional recurrent NN is partially reverse engineered to see what is going on inside the NN as it play Sokoban.
A deep analysis, a lot of manual work has been performed to perform this analysis.
The paper references Mechanistic Interpretability litearture, but the type of probing that is performed in this paper is different.
Nice graphs visualize what goes on inside the RNN as they plan their way through Sokoban. If the authors could make a movie, that would make it even more insightful.
Interestingly, backtracking is found.
The work in the appendix is impressive.

**Questions:**

There are more relevant works on solving Sokoban, for example (I am not one of the authors or related to them)
@inproceedings{shoham2020fess,
  title={The FESS algorithm: A feature based approach to single-agent search},
  author={Shoham, Yaron and Schaeffer, Jonathan},
  booktitle={2020 IEEE Conference on Games (CoG)},
  pages={96--103},
  year={2020},
  organization={IEEE}
}

If, during the course of the review process, you find more and deeper insights, do not hesitate to add them to the final draft.
Please provide videos of the planning process as it progresses inside the NN.

**Ethical Concerns:**

["NO or VERY MINOR ethics concerns only"]

**Final Justification:**

Having read the reviews and viewpoints, and based on the rebuttals, there remain issues concerning clarity of explanation, and significance as a follow up on the original works of MCTS in RNN. Although this is an interesting work, perhaps another conference would be better suited for it. Hence, I have adjusted the score for this paper.

**Limitations:**

Just one game, just one architecture.
This allows an extensive analysis of what goes on inside the NN

**Quality:**

3

**Strengths And Weaknesses:**

Strenghts
- Sokoban is an intriguing puzzle, and is well-chosen as the topic to perform this immensily labor-intensive study
- The insights of what goes on inside the NN as it plans, System 2 reasoning inside a System 1 architecture at work, are impressive
- elaborate appendix, providing extensive insights and study results. Useful for followup work

Weaknesses
- just one application
- just one NN architecture
- how generalizable? Time will tell

Do the strengths offset the weaknesses to the extent that this works should be accepted?
For this reviewer, the depth of the insight do indeed offset the limitations in game and architecture.

---

> ### Author Rebuttal · Authors · 2025-07-31
>
> Thank you so much for your review! We appreciate your understanding that, given our goal to characterize the algorithm learned by the Sokoban agent, we only cover one game/architecture. We hope that this investigation will prove insightful for future analyses.
>
> > There are more relevant works on solving Sokoban
>
> Thank you for providing this work that we missed. We have cited it in our updated draft.
>
> > Please provide videos of the planning process as it progresses inside the NN.
>
> We are glad that the reviewer requested this! Our uploaded supplementary material contains a `videos/` directory that provides visualizations of how the plan of the network to put boxes on the targets evolves while it is playing the game, based on simple linear probes from Table 2 trained to predict future actions. The directory also contains causal intervention videos, and generalization of the network to 3-4x bigger levels based on the actions predicted from the activations.
>
> ---
>
> Please let us know if there are any other questions or clarifications that would be helpful for you!

---

> ### Comment · Reviewer_8R7r · 2025-08-05
> **Thank you for rebuttal**
>
> Having read the reviews and viewpoints, and based on the rebuttals, there remain issues concerning clarity of explanation, and significance as a follow up on the original works of MCTS in NN. Although this is an interesting work, perhaps another conference would be better suited for it.

---

> > ### Author Response · Authors · 2025-08-05
> >
> > We understand that other reviewers raised clarity issues, but we thought that you understood the paper and found “The insights of what goes on inside the NN as it plans” impressive. What specific clarity issues did you find now in retrospect?
> >
> > More importantly, what specifically are the significance issues that you now raise? Other reviewers generally agree that the significance is high, given our novel fairly detailed reverse-engineering of a RNN.
> >
> > Moreover, what work about MCTS in RNNs are you talking about? None of the planning papers (Guez et al., Taufeque et al, Bush et al., this present work) claimed that the RNN is doing MCTS.

---

> > > ### Comment · Reviewer_8R7r · 2025-08-05
> > > **Clarification**
> > >
> > > Thank you for your question for a clarification. Indeed, I see value in your work, it provides insight into how planning can work in NN, and I believe it does build upon the earlier works that. Used CNNs. Hence my score, which is, I believe, the highest among the other reviews. The comment that you refer to should be read in that light: I see value, but I also have questions about how significant, and perhaps a journal would be a better place, allowing for more space that a thorough explanation may need. You may want to link to the other (pure planning) works on Sokoban.

---

### Official Review · Reviewer_fhcS · 2025-07-01

**Clarity:** 1
**Significance:** 2
**Originality:** 3
**Rating:** 2
**Confidence:** 3

**Summary:**

The paper examines a Deep Repeating ConvLSTM (DRC) network trained to plan to solve Sokoban puzzles to understand the underlying mechanism in how it plans. This is accomplished by identifying features relevant to planning, vizualizing convolutional kernels, causal analysis, and ablation studies. The paper finds that causal intervention on the labels it assigns to channels is related semantically meaningful actions in Sokoban, such as movement of boxes or the agent.

The paper itself is not clear in many places and, while there are some nice visualizations, it is missing important background information, methodology for how these visualizations are obtained, and justification for their relation to planning in Sokoban.

**Questions:**

"DRC(3,3)"
- what are the (3,3) parameters?

"Specifically, the solve rate of the DRC improves by 4.7% when the network is given extra thinking time by feeding in the first observation ten times during inference"
- How does this improve performance? What is meant by "thinking"?

What is a "tick"?

What is the purpose of the recurrent architecture? I assume it is to plan. However, are observations given to the DRC recurrently? If so and since the Markov property holds, why?

"Manual inspection of every channel across all layers revealed that most channels are interpretable"
- This appears to be the crux of the paper. How is this done? Are there any other quantifiable metrics that can be used?

"by forward chaining from boxes and backward chaining from targets in parallel in the first few steps."
- What is meant by forward and backward chaining?

**Ethical Concerns:**

["NO or VERY MINOR ethics concerns only"]

**Final Justification:**

I will maintain my rating given the still unclear nature of how planning emerges from the DRC and the subjective nature of channel labeling.

**Limitations:**

The paper relies on manual inspection of the DRC architecture, which is subjective. Furthermore, little information is given in the main text on how this inspection is done.

**Quality:**

2

**Strengths And Weaknesses:**

**Strengths**
The paper proposes several different methods for understanding planning in Sokoban. In particular, given a method to label the role of the channels in the DRC, one can better understand how the plan potentially relates boxes and their goal location and how it considers different plans.

**Weaknesses**
The paper lacks sufficient background and methodology information. The architecture of the DRC is explained somewhat, but how this relates to planning is unclear. It was not clearly stated how the DRC is trained, how planning is incorporated in the training or the architecture, or how data is obtained for training.

The methodology appeared to have manual inspection to accomplish labels for channels, which are then used to do causal analysis. This manual inspection must be subjective and could be biased to support the authors' hypothesis. Therefore, it is unclear if we can trust any subsequent analysis.

While the images, by themselves, appear to show planning behavior, given that they build on features obtained through manual inspection, conclusions drawn from these images become less convincing.

---

> ### Author Rebuttal · Authors · 2025-07-31
>
> Thank you for your review! We answer your questions and objections below:
>
> > The architecture of the DRC is explained somewhat, but how this relates to planning is unclear.
>
> The beauty of this subject is that the architecture of the DRC does not relate to planning by itself! It’s just that this particular instance of a DRC has learned a bidirectional search algorithm (see our reply to XUVs for how the DRC achieves a similar effect to representing lists of nodes to expand). The present work is simply about analyzing the planning algorithm that this particular DRC has learned, not about arguing that the DRC architecture is good at learning to plan.
>
> > It was not clearly stated how the DRC is trained, how planning is incorporated in the training or the architecture, or how data is obtained for training.
>
> As mentioned on page 3, we analyze the open-source DRC(3, 3) network trained by Taufeeque et al. [62] to solve Sokoban. That network followed the training setup of Guez et al. [22], using the IMPALA V-trace actor-critic [18] reinforcement learning (RL) algorithm for 2*10^9 environment steps. Appendix B and C explain the network architecture and training in detail.
>
> Critically, there is no part of the training procedure that explicitly incorporates planning. Instead, the learned planning algorithm emerges from model-free RL.
>
> > This manual inspection must be subjective and could be biased to support the authors' hypothesis. Therefore, it is unclear if we can trust any subsequent analysis.
>
> While the manual inspection was subjective, our subsequent causal analyses were based on quantitative measures of the result of targeted interventions. This is standard scientific process – people come up with hypotheses, then rigorously test them to establish their validity. Please let us know if you think any of our hypotheses are not supported by our experiments to validate them.
>
> > While the images, by themselves, appear to show planning behavior, given that they build on features obtained through manual inspection, conclusions drawn from these images become less convincing.
>
> Since there is an academic consensus that the Sokoban agent exhibits learned planning (Taufeeque et al. [62],  Bush et al. [2], Guez et al. [22], etc.), our paper focuses on how the learned planning algorithm works, rather than arguing for the existence of planning behavior.
>
> We provide several quantitative causal results for key claims.
>
> * Table 2 provides causal intervention scores for the different channel groups, showing that our identified path channels are quantitatively more effective than trained probes in other work at predicting behavior.
> * Figure 3 shows high AUC scores (area under the ROC curve) for the channel activations to predict actions at short and long horizons, which support our interpretation of long and short term path channels.
> * Appendix M demonstrates that using our understanding of the plan extension kernels allows us to modify the network to be capable of solving boards larger than the 10x10 boards that the agent was trained to play on by extending the plan further.
> * Figure 7 shows that ablating the winner takes all mechanism stops the agent from suppressing alternative paths.
>
> What would convince you that the algorithm behaves as explained? Which particular things are unconvincing? We have argued for our understanding of the planning algorithm, and dispute that starting from a manual inspection (as is standard in mechanistic interpretability) renders all subsequent analysis questionable when there is quantitative support for the conclusions.
>
> > what are the (3,3) parameters?
>
> The D, N parameters in DRC(D, N) [from Guez et al. 2017] refer to the Depth of the network, and the Number of repeated applications of the recurrence for every environment step.
>
> > How does this improve performance? What is meant by "thinking"?
>
> It is unclear exactly what this question refers to. This question could mean two things: what do we mean when we claim it “improves performance”? And, in what way does it actually improve the performance?
>
> For the former, performance is the proportion of Boxoban levels (from the validation-medium dataset) which the neural network is able to solve. Solving means putting all four boxes on target squares.
>
> For the latter, we do not fully confidently know how it improves performance, but this paper and previous work [Taufeeque et al. 202?] argue that it does so by giving more computation to the planning algorithm before making irreversible mistakes.
>
> > What is a "tick"?
>
> Given an input, the DRC(3, 3) repeatedly applies the recurrent part of the neural network, three times before deciding on an action for an environment step. We term each of these three times the recurrence gets applied a “tick”, following the original work in Guez et al. 2019.
>
> > What is the purpose of the recurrent architecture? I assume it is to plan. However, are observations given to the DRC recurrently? If so and since the Markov property holds, why?
>
> The architectural choices for the agent were made by Guez et al. in 2017, justified mainly in terms of increased performance in Sokoban.  As such, the architecture was not a free parameter chosen by us – only the DRC(3, 3) was previously demonstrated to have learned planning behaviors, which many other authors have studied. We were interested in the behavior of this neural network, which we undertook to investigate.
>
> That being said, planning is a reasonable guess about the purpose of the architecture. While it is unclear why the architecture (which we did not choose) is provided with the observations at every time step, our understanding of the initialization mechanism and plan extension kernel suggest that it may be necessary in order to establish and stabilize the connection of a path to the target – the plan extension kernels propagate path channel activations along the path, but the initial activation likely comes from adjacency to an important object in the game that needs to be repeated.
>
> > [That most channels are interpretable] appears to be the crux of the paper. How is this done? Are there any other quantifiable metrics that can be used?
>
> We generated hypotheses for these channels subjectively via manual inspection, and at the end we validated our final hypothesis with quantifiable metrics. This process (subjective hypothesis generation with externally observable, quantifiable validation) is explained in Section 3.2.
>
> Table 7 gives concrete labels to 55 out of 96 channels, constituting a majority. Table 2 does causal validation for almost all of these channels – if you change their value, the NN’s output correspondingly changes to the behavior that the channels collectively describe. Figure 3 (middle) shows the AUROC when taking each individual channel’s activations as a classifier to predict future actions, binned by how many steps in the future the action lays, achieving high AUROC scores.
>
> > What is meant by forward and backward chaining?
>
> Forward and backward chaining refer to the process of planning forward from a location (such as the agent or a box) or backward from a target. Here they refer to the two kinds of planning done in bidirectional search (the two lists that reviewer XUVs wrote about):
> * Forward: from a state s1 try all the actions and see what next possible states s2 can be achieved
> * Backward: from a state s2 for each action, figure out which state s1 would have led to s2.
>
> This is a good planning algorithm when both the starting and the goal states are known, as in Sokoban.
>
> > The paper relies on manual inspection of the DRC architecture, which is subjective. Furthermore, little information is given in the main text on how this inspection is done.
>
> As stated earlier and in Section 3.2, we use manual inspection to come up with hypotheses, but we validate them quantitatively. The paper explains our hypotheses, the scoring process, and the quantitative scores they obtained.
>
> The inspection is done by looking at many videos of the agent acting in levels and all of its activations, and coming up with hypotheses that would explain what we see, which we then corroborate with our experiments

---

### Official Review · Reviewer_t3mk · 2025-07-02

**Clarity:** 1
**Significance:** 3
**Originality:** 3
**Rating:** 4
**Confidence:** 1

**Summary:**

The promise of this paper is to scrutinize and explain in familiar terms the mechanisms of an RNN trained via model-free reinforcement learning to play the popular Sokoban game.
The idea is that it will help understand test-time scaling, a new concept observed in particular in some LLMs that enables them to improve their answers when given more compute.

The authors made this particular choice (a Deep Repeating ConvLSTM or DRC playing Sokoban) because it is simpler than LLMs and yet it has been shown to also exhibit test-time scaling.
Decrypting the way this DRC (trained with model-free RL) works, authors give a meaning to the key components of the DRC: channels and kernels from its main convolution layers, and its encoder. This helps them not only understand how the network plays, but also how it forms plans, and how it uses activations of specific channels (path channels) as a kind of value function.

**Questions:**

My main question to the authors is: how do you go from the current version of your paper, to something with technical details that are easier to follow for a wider community of readers, without loosing the depth of your work?

One possible suggestion: develop a higher-level explanation without channels, or kernels, or anything specific to the network architectural details. But first explain what are the steps of the "reasoning" of the network, not with what those steps are made, and do not jump into details right away. For example, if a reader starts by glancing over Figure 1, they will be likely to understand nothing at all. It talks about L1H17 and L1H13 channels, "Linear Plan Extension kernels", "Turn Plan Extension kernels", with activation plots that are difficult to interpret at first...
Another possibility would be to try to explain less but deeper. For example, understanding box-movement or agent-movement channels may not be very important w.r.t. the test-time scaling question, so almost everything not relevant to that central question could be in appendix only.

A few remarks:

- Figure 8 should probably be in the main text.

- In the README of the code, in the line:

pip install -e MambaLens -e farconf -e trained-learned-planners -e gym-sokoban -e stable-baselines3 -e 'learned-planners[torch,dev-local]'

trained-learned-planners should be replaced by train-learned-planners

- On my machine, I tried various ways to install the package but it ended with conflicts, apparently between farconf and trained-learned-planners. But, at least making the effort to share the code is positive, and it is difficult to make packages that are robustly cross-platform, especially when there are many dependencies. Perhaps a docker-type of solution could be something to consider.

- The __MACOSX folder should be removed from the supplementary material.

**Ethical Concerns:**

["NO or VERY MINOR ethics concerns only"]

**Final Justification:**

The rebuttal phase didn't significantly change my opinion on the paper, which is positive with a higher rating than most other reviewers.

I maintain my rating.

**Limitations:**

Yes

**Paper Formatting Concerns:**

No concerns.

**Quality:**

3

**Strengths And Weaknesses:**

This is probably due to the complexity of the attempted task, but I found the paper extremely hard to follow. I am a bit confused because after reading I tend to think that it is good, serious work that has been done, and at the same time I am pretty sure that a large majority of RL experts will not understand it, except the global message of course, when taking it for granted. I don't see a clear path to remedy this situation, and it is not necessarily a huge problem if a group of people who are experts in the interprettion and reverse-engineering process of neural network game-playing agents can benefit from the paper, but the problem is that reviewing it is very difficult for anyone who does not belong to that group of people (like me). I could have said that I was not competent to review the paper, but I did spend a long time trying to understand it, and I think that finding in a short time a replacement reviewer who has the ability to quickly do a better review would have been quite hard, so I chose to make the reviewanyway  and express my opinion after reading the paper, while also signalling that I would probaly need to dedicate an unreasonably long time to that paper to (maybe) make a more confident review. Hopefully not all reviews will be like mine. Most of the time, if I don't follow the paper, I obviously blame it on the authors, but it's not the case here, so I don't think it would be fair to simply rate the paper poorly just because it is very hard to follow. Some topics are intrinsically difficult, and papers with details on game strategies tend to fall in that category. That being said, I think that there is room for improvement, and the paper could be made a bit more accessible to a larger audience.

The paper is very specific: it only interprets one network playing one game, and part of its conclusion is a bit of a stretch that I personally don't believe at this point. I refer to this sentence: "(..) This raises the hope that, if LLM agents are internally performing search, it is possible to find, audit, and correct their goal". A lot of the understanding of the DRC has been made manually by the authors, so I don't see why they believe that their approach already gives hope to interpret the internal search in LLMs. More details on what makes this hope realistic should be given.

But even if the results are not transferable yet to LLMs, the goal of understanding game-playing networks is still very interesting, because it might result in the design of tools with some generic strength to advance our ability to interpret neural networks.

---

> ### Author Rebuttal · Authors · 2025-07-31
>
> Thank you for your very kind review. We see now that our paper is hard to follow, though we tried to make it understandable. The explanation strategy that you propose (first explain the high-level algorithm only, then map it to parts of the neural network) is intriguing, and we will try it for future revisions.
>
> > Figure 8 should probably be in the main text.
>
> It was originally designed to be our Figure 1. But feedback from colleagues said that the paper was difficult to understand this way, because the concrete structures of the neural network that implement each part were not referenced.
>
> > This raises the hope that, if LLM agents are internally performing search, it is possible to find, audit, and correct their goal". A lot of the understanding of the DRC has been made manually by the authors, so I don't see why they believe that their approach already gives hope to interpret the internal search in LLMs. More details on what makes this hope realistic should be given
>
> Our apologies, we cited relevant work, but should have made the connection more explicit.
>
> Our work finds and characterizes the mechanisms for the bidirectional search suggested in Bush et. al [2]. Bidirectional search in LLMs has already been suggested in Anthropic’s “Biology of a Large Language Model” [33]. Given the preliminary evidence for bidirectional search in LLMs, and given our characterization of bidirectional search in Sokoban, our investigation provides hope that similar mechanisms can be found in the LLM setting.
>
> > On my machine, I tried various ways to install the package but it ended with conflicts, apparently between farconf and trained-learned-planners. But, at least making the effort to share the code is positive, and it is difficult to make packages that are robustly cross-platform, especially when there are many dependencies. Perhaps a docker-type of solution could be something to consider.
> Reproducibility and Docker container
>
> We are very pleased that the reviewer was dedicated enough to follow the README and attempt to reproduce the code! And sorry it was a difficult to install mess. The below pip install command with python-3.11 venv should work now. We recommend using uv. We do have a Docker container with the code, but unfortunately, we are unable to offer a docker image in the rebuttal due to NeurIPS’ policies forbidding external links.
>
> ```bash
> uv venv .venv --python 3.11
> source .venv/bin/activate
> uv pip install -e MambaLens -e train-learned-planners -e gym-sokoban -e stable-baselines3 -e 'learned-planners[torch,dev-local]'
> ```

---

> > ### Comment · Reviewer_t3mk · 2025-08-04
> > **Re:**
> >
> > Thank you for the rebuttal.
> >
> > I'd like to maintain my rating.

---

### Official Review · Reviewer_XUVs · 2025-07-06

**Clarity:** 1
**Significance:** 1
**Originality:** 2
**Rating:** 2
**Confidence:** 2

**Summary:**

The paper reverse engineers a recurrent model encoding a policy for solving Boxoban problems. It analyzes different parts of the neural network with techniques from the mechanistic interpretability literature. In the analysis it draws connections to what the models does with known concepts such as a forward model and backtracking.

**Questions:**

If possible, please comment on the points described above where I explain why I couldn't understand the central claims made in the paper.

**Ethical Concerns:**

["NO or VERY MINOR ethics concerns only"]

**Final Justification:**

I thank the authors for answering all my questions. I expect a large number of deep revisions to address all these points in the paper. For example, the response mentions that the bidirectional analogy is looser than an actual bidirectional search algorithm, but not too far from one. I would appreciate a precise description of this bidirectional search being explained in the paper, and with data to back it up.

In the context of a journal, I would read your revised paper and potentially update my recommendation. In the current form, I only have your answers on how you intend to address the issues, but I cannot see them addressed. So it is difficult for me to judge what the paper will be like without reading it. While I appreciate the authors' effort, I will keep my original score.

**Limitations:**

Yes.

**Paper Formatting Concerns:**

No concerns.

**Quality:**

2

**Strengths And Weaknesses:**

The strength of this paper lies in its detailed analysis of a model that is neither too complex nor too simplistic, allowing for a reasonable depth of understanding. The clearly demanded a considerable amount of work applying mechanistic interpretability tools to understand what is encoded in a Boxoban policy. Thinking more generally, understanding some of these opaque models can be a valuable asset for the research community, as it may inspire others to apply these techniques in other application domains.

A key weakness is that the paper itself was quite opaque to me. I understand that people with a stronger background in mechanistic interpretability might think differently. I am an expert on the learning and search components of the paper, but not on the MI techniques. Now that we are clear about my background, I don't see empirical support for many of the claims made in the paper. People from the MI community may think differently, and I will attempt to explain this below.

**Introduction**

The introduction of the paper does not read like an introduction in the sense that it quickly threw me off. I did not follow what was explained in the paragraphs "Representation" and "Algorithm". Take the initial sentence of the first as an example.

*Since the DRC repeats the same computations for each square with limited feedforward depth, [...]*

It is unclear from the context what "square" means.

The second paragraph talks about "path channels", but without defining them, so I am also unable to follow what is stated.

Another issue I had with the introduction is the following claim:

*our work is more complete or more complex than any other related work in mechanistic interpretability*

I am unsure how to judge such a claim; how can one judge completeness and complexity without a clear definition of what they are?

**Methodology**

The interpretability techniques used in the paper are defined briefly, without a clear justification for their use. This is where I realized I am unable to read the paper, because it assumes a level of knowledge I don't have. This feels wrong to me, as this is a general conference and I should be able to follow these ideas, but I am not. If this were a MI conference, I would be fine with accepting that I couldn't understand the paper without reading 3-4 other papers.

While I understand the leftmost image in Figure 3, I don't know what the middle and right plots are showing. For example, what is the average activation shown in the rightmost plot of Figure 3? Are we referring to specific operations within the neural network?

The paper claims that the middle plot shows that the channels collectively predict future movements up to 50 steps ahead. How is that happening? I have no idea how the authors came to this conclusion.

Table 2 has a column titled "Score (%)", but the paper does not explain what this metric is. I can try to infer, and I might be right about it, but the reader shouldn't be left to guess the meaning of Score (%).

The caption of Figure 4 states that the forward and backward plan chains are initialized, but I can't see where this is coming from. How am I supposed to make sense of the heatmaps shown in the plots if the paper doesn't explain them to me?

Something similar happens to Figure 5. How do we know that the "turn right forward kernel" is a turn right kernel? Figure 5 is not even referenced in the text, and I am unsure when to read it.

Figure 7 discusses the WTA not suppressing the right-down path, but I think the authors meant that the right-down path is suppressed with the ablation. Later, the paper states that the WTA ensures that the highest value plans are chosen, but this isn't the case for the example given (Figure 7), as both paths are equally valued. I think that what the WTA does is break ties. The policy learns one of two paths, and the WTA will pick one of them, even if they are equally good.

**Unsupported Claim**

The claim that the model learns to perform a bidirectional search seems wrong to me. There is a chance that the authors are using bidirectional search in a very loose sense, rather than in the classical AI sense, where the search has two open lists and a cost function determines which direction is expanded at a given iteration. To implement such a search, the model would need to implement a queue or a stack (to perform backtracking), and there are works in the literature showing how neural networks can struggle to learn such data structures (e.g., [1]). I am unsure how to reconcile the bidirectional search claim with previous works.

Reference

[1] Joulin, A., & Mikolov, T. (2015). Inferring Algorithmic Patterns with Stack-Augmented Recurrent Nets. ArXiv. https://arxiv.org/abs/1503.01007

**Scholarship Issues**

Maybe this is nitpicking, but it bothers me that the paper doesn't cite papers from the planning literature. For example, it doesn't provide a citation to Alpha Beta pruning or MCTS. When including a citation, it cites Russell & Norvig, which we shouldn't do, as we should cite the original papers. Related to this, I don't understand why the paper even mentions AlphaZero and Alpha Beta pruning; the work is all about a single-agent problem: Sokoban. These algorithms were designed for two-player games, which are not considered at all in this work. MCTS is far from the current state of the art in long-horizon planning problems such as Sokoban, so why bother discussing them?

**Overall Impression**

My review might sound negative, but that may be because I am unfamiliar with the MI literature and couldn't follow the claims made in the paper at all. The work presented may have merit, but it requires a thorough revision before being accepted for publication. I would go further and suggest that this paper should not be submitted to a conference. The authors require significantly more space to convey to the reader the experiments they performed effectively. I want to suggest that the authors consider submitting this work to a journal, where much of the appendix can be incorporated into the main text, and all the techniques used can be explained in a level of detail that will reach a broader community.

---

> ### Author Rebuttal · Authors · 2025-07-31
>
> We’re glad to have an expert in planning reviewing this paper! Thank you so much for your detailed feedback, which we address below.
>
> **Introduction.** Our introduction was intended to cover the generic components of a planning algorithm, and then link them to the properties of the NN that we discover later. Do you mostly have clarifying questions, or would you propose a different structure entirely?
>
> ### Answers to questions:
>
> > It is unclear from the context what "square" means.
>
> In this case, the square is a square in the Sokoban grid.
>
> > The second paragraph talks about "path channels", but without defining them...
>
> We could make this clearer. We define the path channels on line 181 on page 6, as referenced as section 4 in the introduction, but we should have a more succinct presentation earlier.
>
> The DRC(3,3) architecture applies a 3-layer ConvLSTM 3 times (referred as 3 ticks) for each in-game timestep. Each recurrent hidden state $h_d^n$ (layer depth $d$, and “tick” $n$) is an H x W x C tensor representing a Height by Width Sokoban game observation with C channels. We found that a subset of these recurrent hidden state channels describe the path of a box or agent, so we call them the _path channels_.
>
> > how can one judge completeness and complexity without a clear definition of what they are?
>
> We elaborate on this claim in section 6, but could have a more explicit definition. By completeness, we mean how much of the behavior of the network is characterized by the mechanistic analysis offered. By complexity, we mean the sophistication of the characterized behavior, which in some cases can be loosely approximated by parameter count of the model. Some work is a more complete characterization of a simpler system such as modular addition [40, 8, 69, 50, 20, 67]. Other work is a less complete characterization of a more complex system, such as work on LLMs [65, 24, 15, 33, 34, 35, 42]. The present work is a new point on the Pareto frontier, between these two.
>
> ### Methodology
>
> > The interpretability techniques used in the paper are defined briefly...
>
> It is true that the techniques we use are standard in the mechanistic interpretability (MI) literature and might require explanation. We're still optimistic that we can explain the techniques quickly, because they are not complicated. Here is a hopefully better short explanation:
>
> **[H] Feature Identification and Label** This is nothing fancy. We just mean that we manually observe channel activations ($h$) as the agent plays the game, then come up with hypotheses to test what they do. We verify our hypotheses using the methods tagged with [T] explained below.
>
> **[H] Kernel Visualization and Encoder Simplification.** These two methods work together pretty closely – we combine the convolution kernels together (using linear associativity), and then simply visualize them with standard color codings with either the relevant game element colors (for the input encoding), or standard Matplotlib color gradients for visualizing weights.
>
> Equations for our quantitative techniques:
>
> **[H] Direct effect.**
>
> $DirectEffect(i, o) = \max h_d^n[i] * W_io$, where the max is over every square in the grid for every timestep of a level.
>
> **[T] Causal Intervention.** Box path channel causal intervention:
>
> $h’[x, y, c] ← +/-1$ to make the network push a box along the direction of channel c on the square (x, y).
>
> $h’[x, y, c] ← 0$ otherwise.
>
> **[T] Ablation.** We use mean and zero ablation to erase the information contained in some NN activations or weight matrices. These are perhaps the techniques whose rationale requires more explanation, but the procedure is simple enough:
>
> Mean ablation for a channel c with expectation taken over squares (x, y) in the grid and activations sampled from the episodes in Boxoban:
>
> $h’[x, y, c] ← E_{x,y}[h’[x, y, c]]$
>
> Zero-ablation for specific kernel weight in layer d connecting previous channel i to output channel o (for example, zero ablating kernels we’ve identified as WTA kernels):
>
> $$W_d[i, o] ← 0$$
>
> > While I understand the leftmost image in Figure 3...
>
> The average activations in Figure 3 (right) are simply the hidden state ($h$) activations which correspond to the long and short-term down channels mentioned in the caption. $h$ is part of the ConvLSTM block (Fig. 2 and App. B).
>
> > The paper claims that the middle plot shows that the channels collectively predict future movements up to 50 steps ahead. How is that happening? I have no idea how the authors came to this conclusion.
>
> Each line is about the activations of one short-term or one long-term path channel. The value of the activation $h$ at a particular position, combined with a threshold, is the classifier for whether the box will be there $x$ steps in the future. We sweep over possible thresholds to compute the AUC. Then, we observe that the long-term channels have very high AUCs even for positions $x=50$ steps in the future.
>
> We agree that this procedure needs a better explanation.
>
> > Table 2 has a column titled "Score (%)", but the paper does not explain what this metric is. I can try to infer, and I might be right about it, but the reader shouldn't be left to guess the meaning of Score (%).
>
> We can improve the writing next to this figure where it is defined and relabel the column as CI Score for improved clarity – it is the same as the causal intervention (CI) score which is the fraction of times that the causal intervention on the channel results in the agent taking a move congruent with our channel label, in situations where it would have otherwise made a different move.
>
> > How am I supposed to make sense of the heatmaps shown in the plots if the paper doesn't explain them to me?
>
> The path channels are initialized in the sense that they begin with $0$ activation, but that the encoding kernels provide some initial activation which is then propagated by the plan extension kernels. Evidently, our figure 4 caption could be clearer – green in the kernel means that the convolution responds positively to green pixels in the observations.
>
> Notice how the green and red regions are separated at the center of the kernel along the direction of the path channel. Since the hidden state $h$ is initialized to zero at the start, applying these kernels to the observation would add path channel activations along the specific direction (up/down/left/right) from a box or agent or to a target.
>
> For example, the L0O24 kernel activates on a square when the agent is above that square, and is suppressed when the agent is below that square.
>
> > Something similar happens to Figure 5. How do we know that the "turn right forward kernel" is a turn right kernel? Figure 5 is not even referenced in the text, and I am unsure when to read it.
>
> Figure 5 is meant to be read alongside the text next to it, which we will more clearly reference in the text.  As mentioned in the figure, we simply observe the effect of convolving an activation pattern of moving right (leftmost image) with the linear plan extension and right turn kernels (middle column). We see that the linear plan extension kernel extends the right move one further (top right image), and that the turn path extension kernel tells the agent to go up after going right, as explained as the “up move channel” activations.
>
> > Figure 7 discusses the WTA not suppressing the right-down path...
>
> Our figure caption is intended. In the top images (labeled original) the network settles on a single down then right plan, while in the bottom images (the ablation) the network maintains both plan representations.
>
> There appears to be a slight conflation between the value of the plan in the sense of the true reward for following the plan, and the value of the plan in the sense of the network’s learned valuation of the plan. As you mention, moving down then right has the same reward as going right then down. But in the learned valuation, as seen in step 1 for the original network, the network settles on a higher valuation for down then right, which the WTA then reinforces. Ablating the WTA kernels makes the network unable to pick a single path for the box.
>
> > The claim that the model learns to perform a bidirectional search seems wrong to me.... neural networks can struggle to learn such data structures (e.g., [1])
>
> Great point! We use the term in a slightly looser sense than what you describe, but not much looser. Rather than keeping two lists and picking one element from each list, this RNN represents the *set* of already-expanded position-directions, as high values in the path channels for the relevant direction. At each iteration the RNN expands *all* the expanded nodes in parallel, both forwards and backwards. The heuristics that choose which directions to expand from a position, are biased towards re-expanding already expanded directions towards neighbors, so in practice this behaves as if expanding only the frontier. (That is, unless the backtracking signal from §5.3 and App. 5 is present, in which case the frontier recedes and nodes get un-expanded).
>
> When no neighboring states have already been expanded, the heuristics choose 1-2 directions (we have never yet seen 3) to expand.
>
> > Maybe this is nitpicking, but it bothers me that the paper doesn't cite papers from the planning literature.
>
> We’re happy to cite the original papers, but cited Russell and Norvig (R&N) as it was the source for the definitions we used.
>
> > why the paper even mentions AlphaZero and Alpha Beta pruning...
>
> These are just famous search algorithms used as examples. We're interested in a bidirectional search reference beyond R&N if you have one.
>
> > I want to suggest that the authors consider submitting this work to a journal...
>
> Unfortunately, the mechanistic interpretability community has a strong preference for conferences, blog posts, and arXiv submissions. Thanks to your very helpful and engaged feedback, we plan to write a more complete arXiv submission.

---

> > ### Comment · Reviewer_XUVs · 2025-08-05
> > **Thank you!**
> >
> > Thank you very much for answering all my questions. I expect a large number of deep revisions to address all these points in the paper. For example, the response mentions that the bidirectional analogy is looser than an actual bidirectional search algorithm, but not too far from one. I would appreciate a precise description of this bidirectional search being explained in the paper, and with data to back it up.
> >
> > In the context of a journal, I would read your revised paper and potentially update my recommendation. In the current form, I only have your answers on how you intend to address the issues, but I cannot see them addressed. So it is difficult for me to judge what the paper will be like without reading it. While I appreciate the authors' effort, I will keep my original score.

---

### Decision · Program_Chairs · 2025-09-17

**Decision:**

Reject

**Comment:**

This paper presents an attempt to partially reverse-engineer a Sokobon playing neural network, trained purely through model-free RL, to understand the game playing algorithm implemented by the network.

Reviews overall were mixed to negative. The reviewers in general agreed that the authors performed a diligent and overall interesting analysis on an good network/problem combination. Reasons to reject were primarily rooted in multiple reviewers having difficulties following the paper, although two reviewers acknowledged that this may partially be attributed to them not having experience with similar work. Another reviewer did not note clarity issues originally, but later admitted that the paper would benefit from either being heavily revised to make it more accessible, or being published at a different venue. The authors did their best to address many questions during the rebuttal phase, and also provided videos (requested by a reviewer) visualizing the agent's planning process.

This paper is difficult decision. I decided to ignore the review scores in this case and base the meta-review on my own reading of the paper and the reviewer's detailed comments. Clearly, the work put into this analysis was extensive and high-quality. However, I have to agree that the paper as submitted had several clarity issues (documented in good detail by reviewer XUVs) that are likely to leave researchers trying to understand it in depth confused or frustrated. This is unfortunate since the reviewers and I agree that the clarity issues can be fixed (and certain key terms made more precise or changed) with a heavy revision, but we must review the paper as submitted. Therefore, I am recommending a rejection.